# Discovering heterogeneous synaptic plasticity rules via large-scale neural evolution

**Ziyuan Ye**[1*]**, Beichen Huang**[1*]**, Yujie Wu**[2†]**, Guozhang Chen**[3†]**, Jibin Wu**[1,2†]

[1]Department of Data Science and Artificial Intelligence, The Hong Kong Polytechnic University
[2]Department of Computing, The Hong Kong Polytechnic University
[3]National Key Lab. for Multimedia Information Processing, School of Computer Science, Peking University
{ziyuan.ye, beichen.huang, yu-jie.wu, jibin.wu}@polyu.edu.hk,
guozhang.chen@pku.edu.hk

## Abstract

Synaptic plasticity is a fundamental substrate for learning and memory, where different synapse types exhibit distinct plasticity mechanisms. However, how functional behaviors emerge from heterogeneous synaptic plasticity mechanisms remains poorly understood. Here, we introduce a computational framework that harnesses Darwinian evolutionary principles to discover biologically plausible, heterogeneous synaptic plasticity rules within a biologically realistic model of the mouse primary visual cortex. Specifically, we parameterize several key factors related to synaptic plasticity, including presynaptic and postsynaptic spikes, their associated eligibility traces, and neuromodulatory signals. By integrating these factors via a truncated Taylor expansion, we construct a large-scale search space of candidate plasticity rules, with each rule containing over 2.6k optimizable parameters. Each rule is subsequently evaluated on both cross-domain visual task performance and biological validity. Leveraging a multi-objective evolutionary algorithm, we effectively navigate this high-dimensional search space to identify plasticity rules that are both biologically plausible and yield high task performance. We uncover diverse families of high-performing plasticity rules that achieve similar behavioral outcomes despite markedly different mathematical formulations, suggesting that real-world synaptic learning mechanisms may exhibit computational degeneracy. We further show that these biologically plausible rules are not only robust across network scales but also enable few-shot learning, offering a computational explanation for the emergence of innate ability.

## 1 Introduction

A long-standing ambition in both neuroscience and artificial intelligence has been to uncover the mechanisms by which the brain learns to generate intelligent behavior. Synaptic plasticity, as one of the most fundamental mechanisms underlying learning and memory (Bliss & Collingridge, 1993; Markram et al., 2011; McFarlan et al., 2023), has been primarily characterized through experimental studies, revealing a rich repertoire of plasticity mechanisms associated with different types of synapses (Markram et al., 1997; Bi & Poo, 1998; 2001; Woodin et al., 2003; Froemke & Dan, 2002; Kullmann & Lamsa, 2007; Kullmann et al., 2012; D'amour & Froemke, 2015). Nevertheless, the technical challenges inherent in large-scale in vivo experiments have limited our understanding of how these heterogeneous synaptic plasticity mechanisms cooperatively contribute to the emergence of behaviors (McFarlan et al., 2023).

In complement to experimental studies, theoretical modeling approaches aim to reveal the principles underlying synaptic plasticity mechanisms by developing mathematical frameworks that explain how neural activity patterns drive synaptic changes (Hebb, 1949; Bienenstock et al., 1982;

---

*Equal contribution
†Corresponding authors

Oja, 1982; Song et al., 2000; Pfister & Gerstner, 2006; Clopath et al., 2010; Lagzi & Fairhall, 2024; Agnes & Vogels, 2024). Data-driven inference approaches, on the other hand, seek to extract plasticity rules directly from neural recordings or behavioral data using machine learning or statistical methods (Bengio et al., 1991; Stevenson & Koerding, 2011; Robinson et al., 2016; Confavreux et al., 2020; Chen et al., 2023; Mehta et al., 2024; Bell et al., 2024; Kaleb et al., 2024; Confavreux et al., 2025a). However, none of these studies have explored the extensive landscape of heterogeneous plasticity rules in biologically grounded networks, while simultaneously accounting for biological constraints and functional objectives.

In this paper, we investigate the landscape of heterogeneous plasticity rules by asking:

> **Scientific Question**: What mathematical structure and computational principles allow heterogeneous plasticity rules to achieve functional efficacy in realistic cortical circuits?

Our approach leverages evolutionary algorithms to search a broad but interpretable candidate space of heterogeneous plasticity rules on a biologically realistic mouse primary visual cortex (V1) model, where each type of synapse is allowed to employ a distinct learning mechanism. The exploration process is evaluated under multiple objectives that balance task performance with biological plausibility. By simultaneously evaluating a diverse set of candidates within the *population*, our framework enables the study of not just a single "optimal" rule, but a family of rules that span various trade-offs. This includes rules that are potentially very simple yet surprisingly effective, as well as more complex and highly performant ones. As a result, our approach offers a way to explore the rich diversity of plasticity rules that may exist in the brain. The key contributions are:

- **Methodological advances:** (i) We construct an interpretable candidate space of plasticity rules through truncated Taylor expansion, enabling comprehensive enumeration of plasticity rules ranging from simple to complex forms, while maintaining biological interpretability (see Sec. 2.1). (ii) We introduce an evolutionary framework that allows multi-objective optimization on the constructed candidate space for discovering diverse families of plasticity rules (see Sec. 2.2). (iii) We develop several metrics covering task performance and biological constraints, enabling the discovery and systematic evaluation of plasticity rules with varying trade-offs among simplicity, efficiency, task effectiveness, and biological plausibility (see Sec. 2.3).
- **Neuroscientific insights:** (i) Using the proposed framework, we uncovered families of high-performing plasticity rules capable of producing similar memory behavioral outcomes, despite having significantly different mathematical forms (see Sec. 3.2, 4.1 and 4.2). (ii) We show that the evolved plasticity rules discovered by our framework enable few-shot learning, suggesting a potential mechanistic basis for innate abilities (see Sec. 3.3 and 4.3). (iii) We further demonstrate that the explored plasticity rules generalize well beyond their evolutionary settings, maintaining their effectiveness across networks of varying scales (see Sec. 3.4).

## 2 SYNAPTIC PLASTICITY RULE EXPLORATION FRAMEWORK

In this section, we present the proposed multi-objective evolutionary optimization framework, which enables the discovery of biologically plausible synaptic plasticity rules by evolving a large-scale population of candidate rules, as illustrated in Fig. 1.

### 2.1 CONSTRUCTION OF SYNAPTIC PLASTICITY RULE CANDIDATES

Studies have shown that cortical circuits comprise diverse synaptic connections between neuronal populations, which may be governed by distinct plasticity mechanisms (McFarlan et al., 2023). To address this complexity, we build on the mouse V1 cortical model (Billeh et al., 2020; Chen et al., 2022), a biologically grounded spiking recurrent neural network comprising 17 neuron types: four excitatory subtypes from a single class (Exc) and 13 inhibitory subtypes across three classes (Pvalb, Sst, and Htr3a), distributed across six cortical layers (Fig. 1a). Specifically, as shown in Fig. 1b, we consider 289 possible synaptic connection types, corresponding to all pairwise combinations of the 17 neuron types. For each of these synaptic types, we employ a set of commonly identified neural signals in constructing synaptic plasticity rules:

$$\mathcal{T} = \{S_{\text{pre}}, S_{\text{post}}, X_{\text{pre}}, X_{\text{post}}, R\}, \tag{1}$$

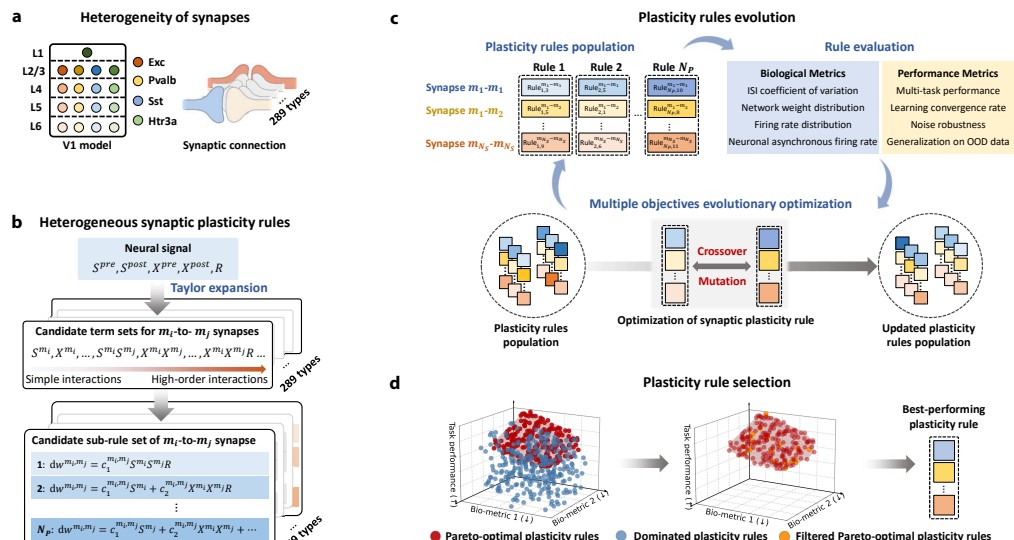

Figure 1: Overview of synaptic plasticity rule exploration framework. **a**. Our framework accounts for synaptic heterogeneity by considering one excitatory (Exc) and three classes of inhibitory synapses (Pvalb, Sst, Htr3a), each highlighted with a distinct color. Light and dark variants further indicate layer-specific subtypes within these classes. **b**. Plasticity rules are constructed from fundamental neural signals, including pre- and postsynaptic spikes, their associated eligibility traces, and the reward prediction error. These signals are systematically combined to generate candidate plasticity sub-rules for each synapse type. **c**. Plasticity rules are evolved through a multi-objective optimization process that simultaneously considers both biological and task performance metrics in a manner that parallels Darwinian evolutionary processes. **d**. The evolutionary search produces a Pareto-optimal subgroup of plasticity rules in the final generation, from which additional constraints are applied to select the filtered population and the overall best-performing rule.

where $S_{\text{pre}}$ and $S_{\text{post}}$ represent pre- and postsynaptic spikes, $X_{\text{pre}}$ and $X_{\text{post}}$ denote pre- and postsynaptic eligibility traces that capture the history of neural activity, and $R$ corresponds to the reward prediction error trace that encodes neuromodulatory signals. These signals serve as fundamental building blocks, enabling the systematic construction of diverse plasticity rules through their additive and multiplicative combinations. It is important to note that the diversity of neuron types can significantly influence the temporal dynamics of both reward prediction error traces (Mohebi et al., 2024) and eligibility traces (Kerlin et al., 2010; He et al., 2015), resulting in heterogeneous plasticity rules across the network. To capture this biological heterogeneity, we use distinct trace decay parameters for different neuron types, as detailed below.

**Reward prediction error trace model**. Neuromodulators, such as dopaminergic signals, typically influence synaptic plasticity in the form of reward prediction errors (Steinberg et al., 2013; Chang et al., 2016; Corkrum et al., 2020; Gershman et al., 2024). Motivated by this finding, we model the neuromodulator signal as a reward prediction error $R$ in our plasticity formulation, with optimizable neuron-type-specific decay time constants $\tau_R$ to capture the potentially heterogeneous temporal profiles of neuromodulation across different neural populations.

In each trial, the neuron population maintains and shares a reward prediction signal, modeled as a simple moving average of recent rewards: $\mathbf{H}_{l_i} = [r_{l_i-1}, r_{l_i-2}, \ldots, r_{l_i-N_{win}}]$, where $N_{win} = 20$ is the window size. This approach reflects the fact that reward signals can persist and influence neuronal activity even after the immediate reward has been received (Hamid et al., 2016). Specifically, the population-shared reward prediction error $\delta_R(l_i)$ for trial $l_i$ can be formulated as:

$$\bar{r}(l_i) = \frac{1}{N_{win}} \sum_{n=1}^{N_{win}} r_{l_i-n}, \tag{2}$$

$$\delta_R(l_i) = r(l_i) - \bar{r}(l_i). \tag{3}$$

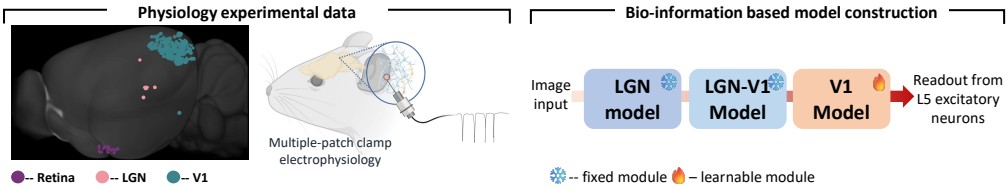

Figure 2: Overview of the biologically realistic mouse V1 model (Billeh et al., 2020; Chen et al., 2022) used in this work.

Eqs. (2)-(3) drives $\delta_R$ close to $0$ when rewards become predictable. Given that neuromodulators signals exhibit distinct dynamics across different synaptic types (Huang et al., 2024b; Mohebi et al., 2024), we model this biological heterogeneity by allowing different neuronal types to maintain distinct reward prediction error traces $R^m$:

$$\frac{dR^m}{dt} = \begin{cases} -\frac{R^m}{\tau_R^m} + \delta_R(l_i) & \text{at trial onset } l_i + 1 \\ -\frac{R^m}{\tau_R^m} & \text{during trial execution,} \end{cases} \tag{4}$$

where $\tau_R^m$ is the time constant for neuron type $m$ that controls the temporal dynamics of the reward signal. Note that the reward prediction error $\delta_R(l_i)$ is only given at the onset of the next trial.

**Eligibility trace model**. Similar to neuromodulatory signals, the eligibility traces of pre- and post-synaptic activities also exhibit heterogeneous temporal dynamics across different neuron types (He et al., 2015). In our model, each neuron type maintains an eligibility trace that decays exponentially over time, with neuron-type-specific time constants to reflect this biological diversity. Specifically, the eligibility trace $X^i$ for neuron $i$ is denoted as

$$X^i(t + \Delta t) = X^i(t) - \frac{\Delta t}{\tau_E^m} X^i(t) + S^i(t), \tag{5}$$

where $\tau_E^m$ refers to the time constant of the eligibility trace for neuron type $m$, and $S^i(t)$ denotes spike occurrence at time $t$.

**Candidate rule set**. We further generate candidate synaptic plasticity terms that act as fundamental building blocks for plasticity rules. Leveraging Taylor expansion's ability to enumerate possible neural signal interactions while preserving biological interpretability, we derive these plasticity terms by expanding the basic neural signals defined in Eq. (1) up to third order. Formally:

$$\mathcal{P} = \left\{ \prod_{j=1}^{q} u_j \ \middle| \ u_j \in \{S_{\text{pre}}, S_{\text{post}}, X_{\text{pre}}, X_{\text{post}}, R\}, \ q \leq 3 \right\}. \tag{6}$$

In practice, redundant terms (e.g., $S_{\text{pre}}^2 = S_{\text{pre}}$ since $S_{\text{pre}}$ is binary) and non-meaningful terms (e.g., $R^2$) are removed. This results in a candidate term set $\mathcal{S} = \{\mathcal{P}^k\}_{k=1}^{N_{\mathcal{P}}}$, where $N_{\mathcal{P}} = 25$, to be used in the subsequent construction of plasticity rules. Each plasticity rule is formulated as a weighted combination of a subset of these candidate terms. Specifically, for a given pair of presynaptic neuronal type $m_{\text{pre}}$, and postsynaptic neuronal type $m_{\text{post}}$, the candidate plasticity rules are defined as:

$$\Delta W^{(m_{\text{pre}}, m_{\text{post}})} = \sum_{k=1}^{N_{\mathcal{P}}} g^k \, c^{k,(m_{\text{pre}}, m_{\text{post}})} \, \mathcal{P}^k, \tag{7}$$

where $c^{k,(m_{\text{pre}}, m_{\text{post}})}$ denotes the coefficient for the $k$-th candidate term specific to synapses connecting neuronal type $m_{\text{pre}}$ to type $m_{\text{post}}$. The variable $g^k \in \{0, 1\}$ is a binary selection gate that determines whether the $k$-th term is active, shared across all synaptic types. According to Eq. (7), the total number of optimizable parameters for each plasticity rule exceeds 2.6K, with the detailed calculation provided in Appendix D. During the weight optimization, Dale's law (Strata et al., 1999) is enforced and hard bounds on the maximum and minimum weight (Gerstner et al., 2014) are implemented to maintain biological plausibility and numerical stability during the weight update. The implementation details are also provided in Appendix D.

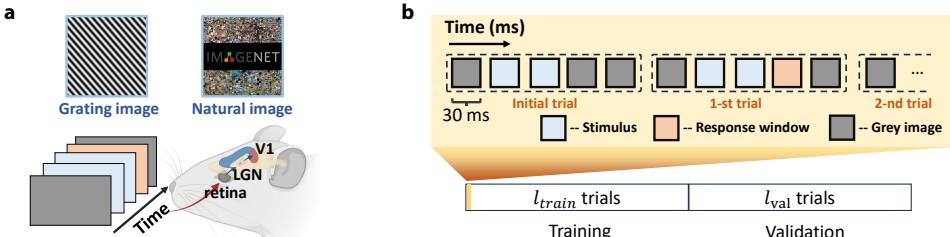

Figure 3: Overview of the multitask experimental setting. **a**. In each evaluation, stimuli are drawn either from gratings or from natural images in ImageNet (not mixed), and are presented to the V1 model. **b**. Each evaluation consists of training trials in the first half, with plasticity rules enabled, followed immediately by validation trials in the second half, with rules disabled.

## 2.2 PLASTICITY RULE EXPLORATION VIA MULTI-OBJECTIVE EVOLUTIONARY OPTIMIZATION

All optimizable parameters of each synaptic plasticity rule are collected in a vector $\boldsymbol{\theta} = \{\boldsymbol{c}, \mathbf{g}, \boldsymbol{\tau}_E, \boldsymbol{\tau}_R\}$, which comprises the plasticity coefficients $\boldsymbol{c}$, binary gates $\mathbf{g}$, and the time constants $\boldsymbol{\tau}_E = \{\tau_E^m\}$ and $\boldsymbol{\tau}_R = \{\tau_R^m\}$ that govern the temporal dynamics of eligibility and reward traces across distinct neuron populations, respectively. This parameterization allows us to formulate synaptic plasticity rule exploration as a multi-objective optimization problem:

$$\text{Minimize } \mathcal{F}(\boldsymbol{\theta}) \in \mathbf{Y}, \boldsymbol{\theta} \in \Omega_\theta, \tag{8}$$

where $\mathcal{F}(\boldsymbol{\theta}) = (f_1(\boldsymbol{\theta}), \ldots, f_{N_o}(\boldsymbol{\theta}))$ represents $N_o$ competing objective functions within the objective space $\mathbf{Y}$, while $\Omega_\theta$ defines the feasible parameter search space. For each candidate synaptic plasticity rule defined by the parameter vector $\boldsymbol{\theta}$, we evaluate its performance by applying it to update the synaptic weights of the V1 model across different cognitive tasks.

As shown in Fig. 2, the input pathway to the V1 model is constructed based on biological data following Billeh et al. (2020). Visual stimuli are first processed by the fixed lateral geniculate nucleus (LGN) and LGN-V1 models before being passed to the learnable V1 model. Model performance is evaluated via readout from the layer 5 excitatory neurons in the V1 model. Model implementation is detailed in Appendix C.

Each plasticity rule undergoes a multi-objective evaluation based on its performance across different cognitive tasks. The evaluation framework incorporates biological plausibility metrics to assess whether both the V1 model's connectivity patterns and neural dynamics align with experimental observations after applying the plasticity rule. The evaluation framework also employs task performance metrics to quantify the plasticity rule's ability to solve the presented cognitive tasks. To explore this complex optimization landscape, we introduce a custom multi-objective evolutionary algorithm, whose design is described in Appendix H. During the evolutionary process, following principles analogous to Darwinian evolution, high-performing plasticity rules are retained and given opportunities to undergo crossover and mutation with other successful plasticity rules, thereby generating novel plasticity rules. Conversely, plasticity rules that exhibit poor performance on cognitive tasks or low biological plausibility are progressively eliminated from the population.

## 2.3 METRICS

To effectively evaluate each plasticity rule across the population, a comprehensive assessment comprising both task performance and biological metrics is employed, as illustrated in Fig. 1c. To maintain computational tractability and ensure a balanced evaluation, the following six objectives are considered in the experiments, including *cross-domain task performance, rule complexity, maximum firing rate, asynchronous firing proportion, firing rate difference and firing rate distribution difference*. We leave the implementation details in the Appendix E.

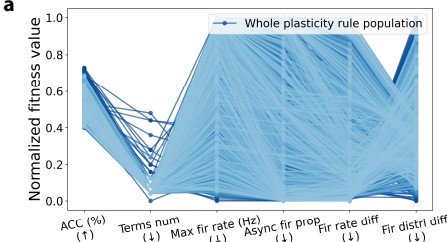 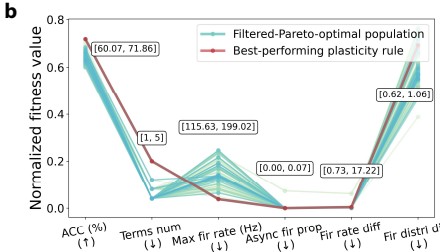

Figure 4: Performance of the filtered plasticity rules after convergence of the population. **a**. Normalized fitness value of 6 objectives for the whole plasticity rule population ($N$=4000). **b**. Normalized fitness value for the filtered Pareto-optimal population ($N$=70). The fitness values are normalized to $[0, 1]$, with original value ranges provided for reference.

| Mathematical formulation of plasticity rules in filtered population | Ratio | Obj-1 (↑) | | Obj-2 (↓) | | Obj-3 (↓) | | Obj-4 (↓) | | Obj-5 (↓) | | Obj-6 (↓) | |
|---|---|---|---|---|---|---|---|---|---|---|---|---|---|
| | | Avg | Best | Avg | Best | Avg | Best | Avg | Best | Avg | Best | Avg | Best |
| $\Delta \mathbf{w} = \mathbf{S_{pre}}$ | **48.57%** | 65.18 | 70.29 | 0.04 | 0.04 | 157.22 | 127.83 | 0.00 | 0.00 | 1.25 | 0.86 | 0.83 | **0.71** |
| $\Delta \mathbf{w} = \mathbf{S_{pre} \cdot X_{post}}$ | 7.14% | 64.54 | 68.43 | 0.04 | 0.04 | 165.10 | 125.96 | 0.00 | 0.00 | 1.16 | 0.87 | 0.91 | 0.86 |
| $\Delta w = S_{post} \cdot X_{pre}$ | 5.71% | 62.50 | 63.21 | 0.04 | 0.04 | 170.62 | 137.43 | 0.00 | 0.00 | 1.04 | **0.84** | 0.91 | 0.85 |
| $\vdots$ | $\vdots$ | $\vdots$ | $\vdots$ | $\vdots$ | $\vdots$ | $\vdots$ | $\vdots$ | $\vdots$ | $\vdots$ | $\vdots$ | $\vdots$ | $\vdots$ | $\vdots$ |
| $\mathbf{\Delta w = X_{post} + S_{pre} \cdot X_{pre} +}$ $\mathbf{S_{post} \cdot X_{pre} + X_{post}^2 + X_{post} \cdot R}$ | 1.43% | 71.86 | **71.86** | 0.20 | 0.20 | 115.63 | **115.63** | 0.00 | 0.00 | 1.76 | 1.76 | 0.98 | 0.98 |

Table 1: Mathematical formulations of plasticity rules ranked by ratio of occurrence in the filtered population and their average performance on two visual change detection tasks. Each formulation may contain multiple rules with identical mathematical form but varying parameters. *Avg*: average fitness across individuals within each formulation; *Best*: optimal fitness within each formulation.

# 3 EXPERIMENTS AND RESULTS

## 3.1 EXPERIMENTAL SETTINGS

**Cross-domain task setting**. Based on the mouse behavioral studies in two visual change detection experiments (Garrett et al., 2020; Siegle et al., 2021), we designed a cross-domain generalization learning (see Fig. 3a). For the natural image change detection, all stimuli are selected from a set of 8 randomly chosen images from the ImageNet dataset (Deng et al., 2009). For the grating image change detection, static grating images are generated with orientations uniformly sampled from the range $[60°, 120°]$ at $0.1°$ precision. The probability of stimulus change between consecutive presentations in two tasks is maintained at 50%. Change detection is signalled when the mean firing rate of the readout excitatory neurons during the response window exceeds a learnable, task-shared threshold $\varphi$. This experiment requires plasticity rules to endow the V1 model with working memory capabilities analogous to those required in 1-back working memory tasks (Owen et al., 2005).

**Evaluation setting**. Each plasticity rule is evaluated independently on each task using a two-phase protocol: 100 training trials (plasticity enabled) followed by 100 validation trials (plasticity disabled). After the population converges, rules undergo a final evaluation using 100 training and 200 testing trials per task to assess average performance across both tasks, with additional testing trials employed to reduce overfitting bias in performance assessment. Our evolution maintained a population of 4,000 plasticity rules across 150 generations, with each individual evaluated by applying its plasticity rule to a V1 model of 3,000 neurons, sampled following the strategy proposed in Chen et al. (2022). Details about the experimental setting and the search space $\Omega_\theta$ can be found in Appendix E.

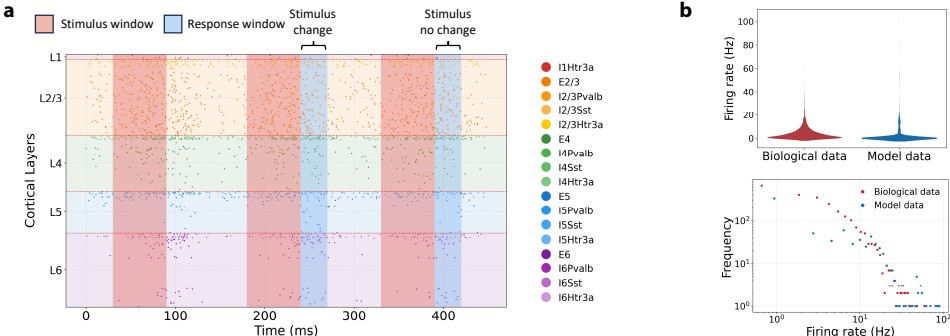

Figure 5: Visualization of spike raster plot and firing distribution of the V1 model optimized by the overall best-performing plasticity rule.

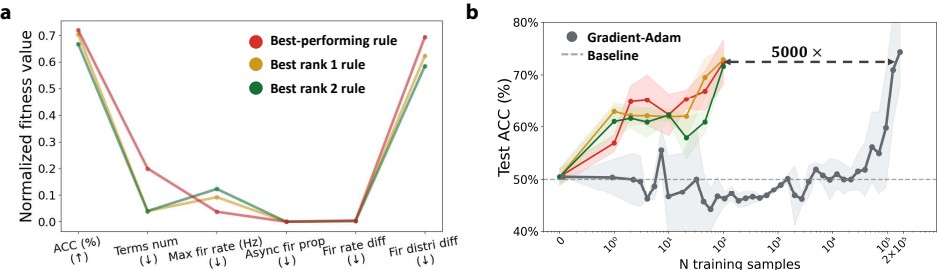

Figure 6: Comparison of performance on visual change detection of grating images between the best individual of three representative rules highlighted in Table 1 and the widely used gradient-based Adam optimizer. **a**. Performance overview. **b**. Mean test accuracy and standard deviation over 5 seeds (200 test trials each).

## 3.2 DISCOVERED PLASTICITY RULES

As shown in Fig. 4, we identified and filtered 70 candidate plasticity rules from the Pareto-optimal population from the last generation based on overall task performance and biological validity criteria. The prevalent mathematical formulations, as demonstrated in Table 1, were ranked and selected according to their ratio of occurrence within the filtered population. We found that most plasticity rules exhibited performance consistent with biological observation. Specifically, the V1 model optimized using these plasticity rules achieves task accuracies comparable to those observed in mice (∼60% for grating change detection (Glickfeld et al., 2013) and ∼73% for natural image change detection (Garrett et al., 2020)). Results were consistent across seeds; one representative seed is shown. The full mathematical formulation table is provided in Appendix F.

In Table 1, the simple presynaptic dependent plasticity rule (*i.e.*, $\Delta w = S_{\text{pre}}$) emerged as the most prevalent formulation, comprising nearly half of the filtered rules. Several reward-free plasticity rules (*e.g.*, $\Delta w = S_{\text{pre}} \cdot X_{\text{post}}$) also succeeded in our reward-required task settings. To further analyze the explored plasticity rules, we selected the overall best-performing plasticity rule from the filtered population, which takes the mathematical form $\Delta w = X_{\text{post}} + S_{\text{pre}} \cdot X_{\text{pre}} + S_{\text{post}} \cdot X_{\text{pre}} + X_{\text{post}}^2 + X_{\text{post}} \cdot R$ (highlighted in red in Fig. 4b and Table 1). The spike raster plot generated by this plasticity rule, as presented in Fig. 5a, illustrates the temporal dynamics of neural activity across cortical layers during both stimulus change and no-change conditions. Fig. 5b demonstrates the comparison of firing rate distributions between biological data (Garrett et al., 2020) and the optimized V1 model, both obtained during the grating change detection task. The analysis reveals that, consistent with the biological data, the optimized V1 model also demonstrates a heavy-tailed firing rate distribution during the grating change detection task.

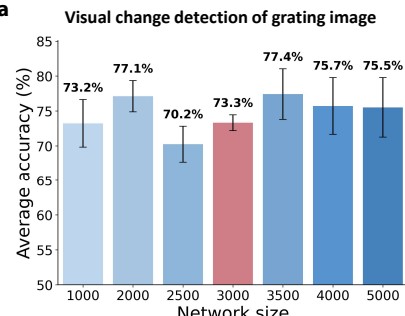 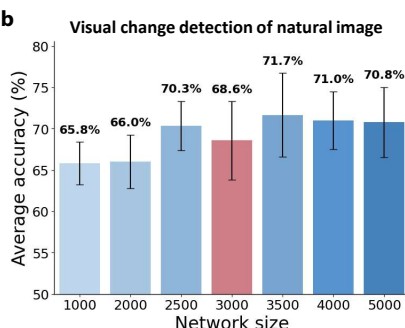

Figure 7: Evaluation of the scalability of the overall best-performing plasticity rule across network sizes. Red bars indicate test accuracy on the V1 model size used during the evolutionary search, while blue bars show test performance on different V1 model sizes. Results represent mean accuracy across 5 independent random seeds, with each evaluated with 200 testing trials. Error bars denote standard deviation across seeds.

### 3.3 DATA EFFICIENCY ANALYSIS

To evaluate the efficiency of the discovered plasticity rules, we compared their performance against a conventional gradient descent (GD) baseline. Specifically, we trained the V1 model on the same visual change detection tasks using the Adam optimizer (Kingma & Ba, 2014) with a surrogate gradient function (Neftci et al., 2019a), serving as a baseline that utilizes global error signals but learns *without evolutionary priors*. To ensure a fair comparison, the biological validity metrics described in Sec. 2.3 were incorporated as regularization terms in the loss function. A detailed training setup is provided in Appendix G.

For benchmarking, we selected three representative rules from our discovered population: the best-performing rules from the top two prevalent formulations (shown in yellow and green in Table 1), and the overall best-performing rule (shown in red). As illustrated in Fig. 6b, we observed a substantial disparity in data efficiency. All three plasticity rules achieved superior detection accuracy within the first 100 training trials, whereas the Adam baseline required nearly 5000× more samples to attain comparable performance levels. In practice, the wall-clock training time for the Adam optimizer is even greater than this factor suggests, due to the additional computational overhead introduced by backpropagation. A similar phenomenon is also observed in visual change detection on natural images (see Fig. S13). The detailed hyperparameter analysis of both Adam and Stochastic Gradient Descent (SGD) is provided in Appendix G.2.

### 3.4 SCALABILITY OF DISCOVERED PLASTICITY RULES

To evaluate the robustness and scalability of our discovered plasticity rules, we tested the overall best-performing plasticity rule across different network scales. Specifically, we varied the V1 model size from 1,000 to 5,000 neurons using the sampling strategy described in Chen et al. (2022). As shown in Fig. 7, the overall best-performing plasticity rule maintained strong performance across different network sizes. This scalability suggests that the discovered plasticity rule captures scale-invariant principles underlying memory formation and retrieval, rather than being overfit to the 3,000-neuron V1 architecture used during the evolutionary optimization. This overall best-performing rule also demonstrates remarkable homeostatic properties under a significantly long time horizon modulation beyond its evolutionary setting (see Appendix F.7).

## 4 DISCUSSION

### 4.1 FUNCTIONALLY EQUIVALENT PLASTICITY RULES

Table 1 demonstrates that high task performance is not restricted to a single plasticity rule. In other words, multiple structurally different computational strategies can generate identical functional behavior, implying a degree of computational degeneracy similar to the recent findings in Confavreux

et al. (2025b). This observation sheds light on why experimental synaptic plasticity studies focusing on the same type of synapse in the visual cortex sometimes report contradictory mechanisms (Lu et al., 2007; Sarihi et al., 2008; Huang et al., 2013), complementing previous explanations based on neuromodulation (McFarlan et al., 2023).

## 4.2 EXPLORED PLASTICITY RULES SUPPORT MEMORY EMERGENCE

Fig. 5 and Fig. S10 reveal that sustained neural firing during delay periods between stimulus windows emerges naturally within the optimized V1 model, which is a canonical neural activity signature of working memory (Fuster & Alexander, 1971; Miyashita, 1988). Notably, *reward-free* plasticity rules relying exclusively on presynaptic activity constitute a significant proportion of the high-performing population (see Table 1) and also result in stable firing dynamics as the reward-required rule (see Fig. S11). This computational efficacy challenges the exclusivity of classical Hebbian coincidence detection and aligns with recent computational modeling in episode memory (Pang & Recanatesi, 2025). It also echoes experimental evidence from hippocampal mossy fiber synapses, where presynaptic activity has been identified as sufficient to support robust memory storage and recall (Vandael et al., 2020; Pelkey et al., 2023; Vandael & Jonas, 2024).

## 4.3 EVOLUTIONARY PRIORS AND THE SYNAPTIC PLASTICITY VIEW OF INNATE ABILITIES

As illustrated in Fig. 6, our comparison with GD is not intended to establish algorithmic superiority, but to provide a possible explanation of the origins of biological learning efficiency. While GD operates as a general-purpose solver requiring extensive data to learn *from scratch*, evolution can embed task-related *inductive biases* into the structure and parameters of the plasticity rules. Consequently, this may enable the plasticity rules, even in a reward-free formulation, to achieve few-shot adaptation. This may explain why the sucking behavior of newborn mice requires maternal pheromone signals to be expressed (Logan et al., 2012). This perspective advances a '*synaptic plasticity view*' of innate behaviors beyond hardwired neural circuitry (Wilmer et al., 2010; Haimson & Mizrahi, 2025): innate capabilities may depend on pre-configured plasticity mechanisms that render specific behaviors accessible with minimal experience, thus bypassing the extensive trial-and-error learning required by acquired capabilities.

## 4.4 LIMITATIONS AND FUTURE WORK

Building on these findings, our framework has several limitations that motivate future extensions. First, while several neural signals are used as building blocks of plasticity rules, incorporating additional variables such as synaptic weights would allow capturing a broader range of possible mechanisms. Second, by grounding our plasticity rules in mouse V1 data and visual processing pathways, we ensure biological fidelity to some extent but limit conclusions about generalization to non-visual sensory modalities, such as auditory data. Third, our framework captures only millisecond-scale plasticity driven by spike timing, without incorporating second-scale mechanisms such as behavioral timescale synaptic plasticity (BTSP) found in hippocampal CA1 (Bittner et al., 2017; Milstein et al., 2021). Fourth, the similarity between the optimized V1 model and biological data is evaluated from a neural representation-based aspect. Future work could incorporate recent advances in dynamics-based similarity metrics (Zhang et al., 2025) to constrain and accelerate the exploration. Addressing these limitations through expanded signal sets, cross-modal validation, multi-timescale integration, and advanced biological constraints represents important future directions.

## 5 CONCLUSION

In this paper, we present a computational framework that employs a multi-objective evolutionary algorithm to discover biologically plausible, heterogeneous plasticity rules within an experimentally grounded mouse V1 model. Our approach uncovers structurally distinct yet functionally equivalent rules, highlighting the role of computational degeneracy in neural robustness. Furthermore, our findings offer potential explanations for the origins of memory and innate abilities, suggesting that efficient learning can emerge from local synaptic dynamics. This work bridges the gap between evolutionary constraints and synaptic diversity, suggesting that the key to biological intelligence may lie in the degenerate yet robust landscape of learning rules.

## ACKNOWLEDGEMENT

This work was partially supported by the National Natural Science Foundation of China (Grant No. 62306259), the Research Grants Council of the Hong Kong SAR (Grant No. C5052-23G, PolyU25216423, and PolyU15217424), The Hong Kong Polytechnic University (P0058445). This work was also supported by the National Natural Science Foundation of China (NSFC) under Grant No. 62576011.

## ETHICS STATEMENT

This work adheres to the ICLR Code of Ethics and addresses several ethical considerations relevant to computational neuroscience and artificial intelligence research:

**Biological data usage**. Our research builds upon publicly available mouse V1 cortical data from the Allen Brain Atlas (Billeh et al., 2020), which was collected under appropriate institutional ethical approvals. We do not conduct new animal experiments and rely exclusively on established, ethically-sourced datasets.

**Computational resource considerations**. While our evolutionary framework requires substantial computational resources (8×A6000 GPUs), we have designed efficient algorithms based on EvoX (Huang et al., 2024a) to minimize environmental impact through batched evaluation and gradient-free optimization. The computational efficiency gained enables broader scientific exploration while reducing overall resource consumption compared to conventional approaches.

**Potential applications and dual use**. The discovered plasticity rules could inform the development of more efficient neuromorphic computing systems and brain-inspired AI. While these advances may have beneficial applications in assistive technologies and computational efficiency, we acknowledge that any powerful learning algorithm could potentially be misused. However, our work focuses on fundamental scientific understanding rather than specific applications, and the biological constraints inherent in our approach naturally limit potential for misuse.

**Research integrity**. All experimental protocols, data processing steps, and evaluation metrics are fully documented to ensure transparency and reproducibility. Our multi-objective evaluation framework explicitly incorporates biological plausibility constraints to prevent the discovery of biologically implausible but computationally expedient solutions.

## REPRODUCIBILITY STATEMENT

We have made extensive efforts to ensure the reproducibility of our results:

**Computational framework**. Complete implementation details of our multi-objective evolutionary algorithm are provided in Appendix F, including the custom selection operators, reproduction mechanisms, and evaluation procedures. The framework is implemented using JAX and the EvoX library, with specific and hardware requirements detailed in Appendices E and H.

**Model specifications**. The V1 model architecture, connectivity patterns, and neuronal parameters are fully specified in Appendix A, building upon the publicly available Allen Brain Atlas data (Billeh et al., 2020) and following the architecture modification in Chen et al. (2022). All modifications to the original GLIF$_3$ model (Teeter et al., 2018) for plasticity compatibility are explicitly documented.

**Experimental protocols**. Detailed experimental settings, including stimulus generation, task protocols, evaluation metrics, and hyperparameter ranges, are provided in Appendices D and E. The visual change detection paradigm follows established behavioral protocols (Garrett et al., 2020; Siegle et al., 2021).

**Baseline comparisons**. The gradient-based Adam optimizer baseline implementation, including surrogate gradient functions and training procedures, is fully described in Appendix G to enable direct replication of comparative results.

**Statistical analysis**. All reported results include appropriate statistical measures (means, standard deviations, sample sizes) across 5 random seeds. The evolutionary search was conducted with a

population of 4,000 rules over 150 generations, with final evaluations performed using 5 independent seeds and 200 test trials each.

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

# A  SUMMARY OF NOTATIONS

This section provides a summary of the important notations and variables that are frequently used throughout the paper in Table S2.

Table S2: Summary of important notations

| Notation | Description |
|---|---|
| **General Indices** | |
| pre, post | Indices denoting presynaptic and postsynaptic quantities, respectively. |
| $(m_{\text{pre}}, m_{\text{post}})$ | A pair of presynaptic and postsynaptic neuronal types uniquely identifies a certain synapse type. |
| **Plasticity Signals** | |
| $S$ | Binary indicator for spike occurrence (e.g., $S_{\text{pre}}, S_{\text{post}}$). |
| $X$ | Eligibility trace capturing the history of neural activity (e.g., $X_{\text{pre}}, X_{\text{post}}$). |
| $R$ | Reward prediction error trace encoding neuromodulatory signals. |
| **Plasticity Parameters** | |
| $c$ | Learnable coefficient for a specific plasticity term (e.g., $c^k$). |
| $\tau_E^m$ | Time constant for the eligibility trace of neuron type $m$. |
| $\tau_R^m$ | Time constant for the reward trace of neuron type $m$. |
| $\mathcal{P}$ | Set of candidate plasticity terms derived from Taylor expansion. |
| $\Delta W$ | Synaptic weight update value (plasticity rule). |
| **Network & Neuron Model** | |
| $W_{\text{LGN}}$ | Fixed synaptic weights from the LGN model to the V1 model. |
| $\tau_{\text{syn}}$ | Synaptic time constant governing current decay. |

# B    RELATED WORK

## B.1    MODELING PLASTICITY MECHANISM

Explaining the synaptic plasticity mechanism via mathematical formulation has been a main theme in computational neuroscience. Initial theoretical frameworks for synaptic plasticity, including BCM theory (Bienenstock et al., 1982) and Oja's rule (Oja, 1982), were largely based on Hebbian assumptions (Hebb, 1949). Later experimental evidence revealed that the precise timing of spikes is crucial for synaptic plasticity (Bi & Poo, 1998; Zhang et al., 1998; Debanne et al., 1998; Woodin et al., 2003). This led to the development of a rich body of spike timing-dependent plasticity (STDP) models (Song et al., 2000; Kempter et al., 1999; Pfister & Gerstner, 2006; Clopath et al., 2010). More recently, spike timing-based plasticity models have established connections with network properties, as well as network learning and memory characteristics, offering an integrated view of how synaptic plasticity shapes neural computation (Zenke et al., 2015; Illing et al., 2021; Payeur et al., 2021; Eckmann et al., 2024; Brito & Gerstner, 2024; Agnes & Vogels, 2024).

Despite remarkable progress in understanding individual synaptic plasticity mechanisms in isolation, experimental studies suggest that multiple types of synaptic plasticity often operate synergistically (D'amour & Froemke, 2015; El-Boustani et al., 2018). These processes commonly involve various neuromodulators (Seol et al., 2007; Huang et al., 2013; McFarlan et al., 2023; Park et al., 2025), which collectively shape network dynamics and functional behavior. However, technical limitations inherent to large-scale in vivo experiments have hindered a comprehensive understanding of how these heterogeneous synaptic plasticity mechanisms interact.

## B.2    INFERRING PLASTICITY MECHANISM

In addition to modeling plasticity rules, an alternative approach to understanding plasticity mechanisms involves inferring their mathematical formulations directly from neural or behavioral data using data-driven methods. Predicting plasticity rules from spike train data, in particular, represents an intuitive and powerful means of uncovering the underlying synaptic mechanisms (Stevenson & Koerding, 2011; Linderman et al., 2014; Robinson et al., 2016; Ghanbari et al., 2017; Wei & Stevenson, 2021; Mehta et al., 2024). Previous studies have also derived synaptic plasticity rules by examining changes in neuronal firing distributions observed before and after learning (Lim et al., 2015; Chen et al., 2023). Alternatively, some studies explore plasticity mechanisms by identifying how synaptic modifications enable networks to achieve desired network dynamics and computational functions (Confavreux et al., 2020; 2023). Complementarily, another class of studies optimizes synaptic rules to endow networks with the capability for successful cognitive task performance or behavioral data fitting (Najarro & Risi, 2020; Ashwood et al., 2020; Jordan et al., 2021; Tyulmankov et al., 2022; Shervani-Tabar & Rosenbaum, 2023; Miconi, 2023; Mehta et al., 2024; Confavreux et al., 2025a;b).

Although some of the aforementioned studies have employed evolutionary methods or meta-learning frameworks to explore plasticity mechanisms (Najarro & Risi, 2020; Jordan et al., 2021; Tyulmankov et al., 2022; Confavreux et al., 2023; Miconi, 2023; Shervani-Tabar & Rosenbaum, 2023; Confavreux et al., 2025a), these approaches suffer from several limitations. First, they have largely been restricted to small artificial networks or simplified biological models, which may limit their ability to incorporate biological constraints and capture the full complexity of biological systems. Second, most current methods are restricted to a small, manually defined space of plasticity rules, typically spanning from a uniform rule applied across all synapses to limited forms of heterogeneity that distinguish only between excitatory and inhibitory synapses. Third, prior studies have primarily been limited to single-task settings or have considered only a single biological objective. These limitations hinder comprehensive exploration of the broader landscape of heterogeneous plasticity rules, as well as the systematic identification of synaptic plasticity mechanisms that are both biologically plausible and functionally relevant.

# C   MODEL CONSTRUCTIONS OF LGN AND V1

Our computational framework employs a hierarchical visual processing architecture comprising a fixed LGN preprocessing stage and an adaptive V1 cortical network. Specifically, the LGN model transforms visual stimuli through 17,400 spatiotemporal filters, while the V1 network, built on experimentally validated Allen Brain Atlas parameters (Billeh et al., 2020), serves as the substrate for plasticity rule discovery across 17 distinct cell types and cortical layers.

## C.1   LGN MODEL AND LGN-TO-V1 MODEL

Visual stimuli undergo preprocessing through an LGN model that qualitatively simulates the computational functions of both retinal and LGN processing stages (Billeh et al., 2020). The model architecture incorporates 17,400 spatiotemporal filters designed to replicate the response characteristics of mouse LGN neurons to visual inputs (Durand et al., 2016). Each filter generates positive-valued outputs representing the firing rates of individual LGN neurons.

The preprocessing pipeline begins with converting visual input pixels to grayscale, followed by normalization to the interval of intensity $[-\text{Int}, \text{Int}]$ where $\text{Int} = 2$ in our setting. The processed LGN outputs are subsequently transformed into external current injections for the V1 model through a fixed biological-data-based LGN-to-V1 connection weight. The whole LGN to V1 pathway can be represented as:

$$I_{\text{sti}} = \mathbf{W}_{\text{LGN}} \cdot \text{LGN}(G_{\text{Int}}). \tag{9}$$

where $G_{\text{Int}}$ denotes the intensity-normalized image inputs within the range $[-2, 2]$ following the settings in Chen et al. (2022). Note that the LGN model $\text{LGN}(\cdot)$ and LGN-to-V1 model $\mathbf{W}_{\text{LGN}}$ remain fixed during the whole optimization process, as we only focus on the plasticity within the V1 model.

## C.2   V1 MODEL

**Neuron models**. Our computational framework builds upon the point-neuron implementation of the biologically realistic V1 model developed by Billeh et al. (2020). To ensure compatibility with plasticity learning rules, we substituted the discrete membrane potential reset mechanism following spike generation with a continuous voltage reduction term $z_j(t)(v_{th} - E_L)$, where the spike indicator $z_j(t) = 1$ during firing events of neuron $j$ at time $t$, and $z_j(t) = 0$ otherwise. Here, $v_{th}$ represents the spiking threshold and $E_L$ denotes the resting membrane potential. This modification preserves the essential neural dynamics while enabling synaptic plasticity rule-based weight updates.

The temporal evolution of the spiking model follows the modified $\text{GLIF}_3$ model (Teeter et al., 2018), which can be represented as:

$$\begin{aligned}
v_j(t + \Delta t) = \ &\beta v_j(t) - z_j(t)(v_{th} - E_L) \\
&+ \frac{1 - \beta}{C}\left( I_j^e(t + 1) + \sum_m I_j^m(t + 1) + \mathsf{g}E_L + I_j^{syn}(t) \right)
\end{aligned} \tag{10}$$

$$z_j(t) = H(v_j(t) - v_{th}) \tag{11}$$

$$I_j^e(t) = \sum_i W_{\text{LGN}}^{i,j} \text{LGN}(G_{\text{Int}})_i(t) \tag{12}$$

where $C$ represents neuron capacitance, $I_j^e$ denotes external current input, $I_j^m$ refers to the post-spike current, $I_j^{syn}$ represents synaptic current, g specifies membrane conductance, and $v_{th}$ indicates the spiking threshold. The LGN-to-V1 connectivity is mediated by the weight matrix $\mathbf{W}_{LGN}$. The decay factor $\beta = e^{-\Delta t / \tau}$ incorporates the membrane time constant $\tau$, while $\Delta t = 1$ ms defines the resolution of the simulation time step. The Heaviside step function $H(\cdot)$ governs spike generation dynamics.

To model the neuronal refractory period, we implemented a simplified mechanism where $z_j(t)$ remains fixed at 0 following each spike for a brief refractory period determined by neuron type. Postsynaptic spike current dynamics are governed by:

$$I^m(t + \Delta t) = f^m I^m(t) + z(t) \Delta I^m; \quad m = 1, \ldots, N_{asc} \tag{13}$$

where the multiplicative decay constant $f^m = \exp(-k^m \Delta t)$ and additive term $\Delta I^m$ characterize the after-spike current properties. Our implementation supports $m = 1$ or 2 current types. Biophysical parameters are derived from experimental measurements of 111 neurons in the Allen Brain Atlas database (Durand et al., 2016; Billeh et al., 2020), encompassing neuron capacitance $C$, conductance $g$, resting potential $E_L$, refractory period duration, and the amplitudes $\Delta I^m$ and decay constants $k^m$ for both after-spike current types.

**Synaptic connectivity and dynamics**. The V1 network architecture incorporates experimentally-derived connectivity patterns that define probabilistic synaptic connections between neuron populations. Connection probabilities for all pairwise combinations among the 17 characterized cell classes are specified according to experimental measurements (Billeh et al., 2020), with data organized in a comprehensive connectivity matrix. Grid entries indicating unknown values represent uncharacterized connection types in the experimental dataset.

The baseline connection probabilities reflect measured synaptic contact frequencies for neuron pairs positioned within 75-$\mu m$ horizontal intersomatic distances. To incorporate spatial connectivity constraints, these probabilities are modulated by an exponentially decaying factor that scales with horizontal distance between neuronal somata following Chen et al. (2022) Fig.1D. This distance-dependent scaling preserves the statistical properties observed in experimental connectivity data while accounting for spatial organization principles in cortical circuits.

Synaptic transmission delays are distributed uniformly within the interval [1,4] ms, derived from experimental measurements presented in Fig. 4E of Billeh et al. (2020) and discretized to match the 1 ms integration time step. The postsynaptic current dynamics for neuron $j$ follows first-order kinetics:

$$I_j^{syn}(t + \Delta t) = e^{-\frac{\Delta t}{\tau_{syn}}} I_j^{syn}(t) + \Delta t \cdot e^{-\frac{\Delta t}{\tau_{syn}}} C_j^{rise}(t), \tag{14}$$

$$C_j^{rise}(t + \Delta t) = e^{-\frac{\Delta t}{\tau_{syn}}} C_j^{rise}(t) + \sum_i W_{V1}^{i,j} z_i(t) \frac{1}{\tau_{syn}}, \tag{15}$$

where $\tau_{syn}$ represents the synaptic time constant, $W_{V1}^{i,j}$ denotes the recurrent connection weight from presynaptic neuron $i$ to postsynaptic neuron $j$ within V1, and $z_i(t)$ indicates spike occurrence in the presynaptic neuron. The synaptic time constants $\tau_{syn}$ are cell-type specific, reflecting the diverse kinetic properties of synaptic transmission between different neuronal populations as characterized in Billeh et al. (2020).

**Network initialization**. All state variables, including spike indicators and membrane potentials, are initialized to zero at simulation onset. The initial configurations of both feedforward weights $\mathbf{W}_{LGN}$ and recurrent connectivity weights $\mathbf{W}_{V1}$ are established according to the empirical values reported in Billeh et al. (2020), providing a biologically realistic starting point for network dynamics and subsequent plasticity-driven modifications.

# D DETAILS ABOUT PLASTICITY RULES AND WEIGHT UPDATES

## D.1 PLASTICITY RULE CANDIDATES TERMS

Here we present all the possible plasticity rule terms that may exist in Eq. 7 in the main manuscript. Each plasticity rule candidate is governed by a binary selection indicator $g$ and a learnable coefficient $c$. This term set includes the *first-order terms:*

$$g^1 \cdot c^1 \cdot S_{\text{pre}}, \ g^2 \cdot c^2 \cdot S_{\text{post}}, \ g^3 \cdot c^3 \cdot X_{\text{pre}}, \ g^4 \cdot c^4 \cdot X_{\text{post}}, g^5 \cdot c^5 \cdot R;$$

the *second-order terms:*

$$g^6 \cdot c^6 \cdot S_{\text{pre}} \cdot S_{\text{post}}, \ g^7 \cdot c^7 \cdot S_{\text{pre}} \cdot X_{\text{pre}}, \ g^8 \cdot c^8 \cdot S_{\text{pre}} \cdot X_{\text{post}}, \ g^9 \cdot c^9 \cdot S_{\text{pre}} \cdot R,$$
$$g^{10} \cdot c^{10} \cdot S_{\text{post}} \cdot X_{\text{pre}}, g^{11} \cdot c^{11} \cdot S_{\text{post}} \cdot X_{\text{post}}, \ g^{12} \cdot c^{12} \cdot S_{\text{post}} \cdot R,$$
$$g^{13} \cdot c^{13} \cdot X_{\text{pre}}^2, \ g^{14} \cdot c^{14} \cdot X_{\text{pre}} \cdot X_{\text{post}}, \ g^{15} \cdot c^{15} \cdot X_{\text{pre}} \cdot R, \ g^{16} \cdot c^{16} \cdot X_{\text{post}}^2,$$
$$g^{17} \cdot c^{17} \cdot X_{\text{post}} \cdot R;$$

and the *third-order terms:*

$$g^{18} \cdot c^{18} \cdot X_{\text{pre}} \cdot S_{\text{post}} \cdot R, \ g^{19} \cdot c^{19} \cdot S_{\text{pre}} \cdot X_{\text{post}} \cdot R, \ g^{20} \cdot c^{20} \cdot X_{\text{pre}}^2 \cdot R,$$
$$g^{21} \cdot c^{21} \cdot X_{\text{post}}^2 \cdot R, \ g^{22} \cdot c^{22} \cdot S_{\text{pre}} \cdot S_{\text{post}} \cdot R, \ g^{23} \cdot c^{23} \cdot X_{\text{pre}} \cdot X_{\text{post}} \cdot R,$$
$$g^{24} \cdot c^{24} \cdot S_{\text{post}} \cdot X_{\text{post}} \cdot R, \ g^{25} \cdot c^{25} \cdot S_{\text{pre}} \cdot X_{\text{pre}} \cdot R,$$

where $g$ serves as a binary indicator, and $c$ denotes the coefficient for candidate. Among the 25 candidate plasticity terms presented above, 16 terms depend only on neuron types, 8 terms are synapse-type specific, and 1 term represents the global reward prediction error. Two additional decay factors govern the neuron-type-dependent eligibility trace and reward prediction error trace.

**Optimizable parameter analysis.** The total number of optimizable parameters for each of the plasticity rules in our framework can be decomposed as follows:

$$n_{\text{optimizable}} = 18 \times n_{\text{neuron types}} + 8 \times n_{\text{synapse type}} + n_{\text{gate}} + 2. \tag{16}$$

For our V1 model implementation following Billeh et al. (2020), which contains 17 distinct neuron types and thus $17^2 = 289$ possible synapse types, combined with our 25 candidate plasticity terms, the total number of optimizable parameters becomes:

$$n_{\text{optimizable}} = 18 \times 17 + 8 \times 17^2 + 25 + 2 = 2645. \tag{17}$$

This formulation accounts for several parameter categories:

- **Neuron type specific coefficients**. The first term ($18 \times n_{\text{neuron types}}$) corresponds to 16 plasticity rule coefficients that depend only on either presynaptic or postsynaptic neuron type, encompassing most first-, second-, and third-order terms as well as the 2 neuron-type-dependent decay factors.
- **Synapse type specific coefficients**. The second term ($8 \times n_{\text{synapse type}}$) represents coefficients that depend on synapse types (*i.e.*, both presynaptic and postsynaptic neuron types).
- **Binary selection gates**. The term $n_{\text{gate}}$ represents the number of binary indicators $g$ for each candidate term in the plasticity rule set.
- **Additional parameters**. The constant term 2 accounts for: (i) the global coefficient for the reward prediction signal $R$ in first-order terms, which is shared across all synaptic types; and (ii) the task-shared learnable readout threshold $\varphi$ used in the decision-making process.

Note that among the first-order terms, the reward prediction signal $R$ has only a single coefficient and gate parameter that can be optimized globally, rather than being specific to individual synaptic connections, reflecting its role as a neuronal population-level learning signal.

## D.2 WEIGHT UPDATE CONSTRAINTS

We enforce Dale's law (Strata et al., 1999) during plasticity rule based weight updates to maintain biological plausibility and numerical stability. Dale's law, which requires neurons to release identical neurotransmitters across all synapses, ensures that synaptic polarity remains invariant throughout network optimization. Moreover, due to physical constraints in biological synapses, we consider the synapse type-specific thresholds to prevent weights from becoming excessively large, which can be treated as hard bounds during the network optimization (Gerstner et al., 2014). We set these bounds at 1.5 times the maximum biological values (Billeh et al., 2020) for each synapse type. We also implement an adaptive scaling mechanism to prevent excessive weight changes that could destabilize network dynamics. The weight updates are constrained by a threshold proportional to the current weight magnitude, where the proportion is set to 0.01. This approach ensures that weight modifications remain proportional to existing connection strengths, preventing dramatic changes that could disrupt established connectivity patterns.

# E    EXPERIMENTAL DETAILS

## E.1    DETAILED IMPLEMENTATION OF THE EVALUATION METRICS

1. **Cross-domain task performance**, measured by cross-domain task average accuracy, assesses whether the plasticity rule generalizes across different stimulus types rather than specializing to a particular type. In this work, we measure it as the average validation accuracy from independent training sessions using two distinct input domains (grating images and natural images). The accuracy is computed as the proportion of correct responses, including hits during change trials and correct rejections during no-change trials.

2. **Rule complexity** assesses the number of terms within each plasticity rule, with preference given to simple formulations when comparable performance is achieved. In our implementation, we quantify rule complexity by counting the number of active terms in the plasticity rule, determined by the binary selection indicators $g$ of Eq. (7).

3. **Maximum firing rate** monitors peak neuronal firing rates during the cognitive task after the plasticity rule optimization has been done, with rules maintaining firing rates below 30 Hz considered biologically optimal (Niell & Stryker, 2008). In our implementation, we record the maximum firing rate across all neurons in the network during the validation phases.

4. **Asynchronous firing proportion** evaluates population-level synchronous firing, requiring that simultaneously active neurons constitute less than 1% of the total population at any time point (Lennie, 2003). In our implementation, we calculate the proportion of neurons firing simultaneously at each time step during the validation phase, and identify the maximum proportion.

5. **Firing rate difference** measures the discrepancy between the network's population firing dynamics during cognitive tasks, after the plasticity rule optimization has been done, and the mean firing rate data observed in mice performing analogous cognitive tasks (Siegle et al., 2021; de Vries et al., 2023). In our implementation, we compute the absolute difference between the average firing rate of the network during validation phase and the experimentally recorded mean firing rate from mice.

6. **Firing rate distribution difference** quantifies the distributional differences between the network's population firing dynamics during cognitive tasks, after the plasticity rule optimization has been done, and the firing rate distributions recorded from mice performing analogous cognitive tasks (Siegle et al., 2021; de Vries et al., 2023). In our implementation, we compute the Wasserstein distance between the normalized firing rate distributions of the network during validation phase and the experimentally recorded firing rate distributions from mice.

## E.2    CROSS-DOMAIN TASK SETTING

The original behavioral protocol (Siegle et al., 2021; de Vries et al., 2023) required mice to perform sequential discrimination on static natural images or grating images (250 ms presentation duration) interspersed with inter-stimulus grey screen intervals (500 ms), determining whether consecutive stimuli are identical or different. To enable large-scale population search, we implemented 60 ms stimulus presentations and 90 ms inter-stimulus grey intervals (Fig. 3b). Each trial sequence initiated with a 30 ms pre-stimulus delay. Network responses are evaluated within a 30 ms response window beginning 60 ms post-stimulus onset.

## E.3    SEARCH SPACE SETTING

The search spaces of different parameters in each of the plasticity rules are set as follows: Coefficient of plasticity candidate $c \in [-1, 1]$, binary selection indicator $\mathbf{g} \in \{0, 1\}$, decay factor of eligibility trace $\tau_E \in (0, 150]$, and decay factor of reward prediction error $\tau_R \in (0, 150]$. The task-shared learnable readout threshold $\varphi \in [0, 10]$ Hz, following mouse cortical pyramidal neuron firing rates where most neurons fire below 10 Hz (Buzsáki & Mizuseki, 2014).

To gauge the complexity of this search space, we conducted a preliminary analysis by evaluating $3,000$ randomly sampled solutions from the defined search space. The statistical distribution of the

Table S3: Statistics of 3,000 randomly sampled solutions across six objectives.

| Metric | Mean | Std | Median | Best | Worst |
|---|---|---|---|---|---|
| Obj 1: Cross-domain task performance ($\uparrow$) | 0.503 | 0.016 | 0.500 | 0.685 | 0.440 |
| Obj 2: Rule complexity ($\downarrow$) | 0.501 | 0.097 | 0.520 | 0.200 | 0.840 |
| Obj 3: Maximum firing rate ($\downarrow$) | 437.21 | 122.96 | 500.00 | 99.33 | 500.00 |
| Obj 4: Asynchronous firing proportion ($\downarrow$) | 0.754 | 0.408 | 1.000 | 0.000 | 1.000 |
| Obj 5: Firing rate difference ($\downarrow$) | 139.13 | 93.66 | 144.87 | 0.62 | 271.71 |
| Obj 6: Firing rate distribution difference ($\downarrow$) | 0.612 | 0.201 | 0.610 | 0.144 | 1.253 |

six objectives is summarized in Table S3. The results demonstrate that the search space is highly non-trivial. Judging by the mean and median values, the vast majority of randomly sampled plasticity rules fail the task. While the best-observed values (Maximum for task performance, minimum for constraints) indicate that high-performing candidates exist within the theoretical bounds, these statistics are computed independently for each objective. Notably, a solution that achieves a good score on one metric (e.g., good accuracy) may perform poorly on others (e.g., extremely high firing rate).

Our evolution framework and model implementation are built on Jax (Bradbury et al., 2018). All the experiments were conducted on 8 ×A6000 GPUs. The simulations were conducted with 1 ms temporal resolution.

### E.4 BEHAVIOR DATA ESTIMATION FOR VISUAL CHANGE DETECTION

The behavioral accuracy of visual change detection on *natural images* is estimated from Fig. 1I of Garrett et al. (2020) when familiar images were presented to mice. The behavioral accuracy of visual change detection on *grating images* is estimated from Fig. 3A of Glickfeld et al. (2013) when the orientation difference was 5 degrees.

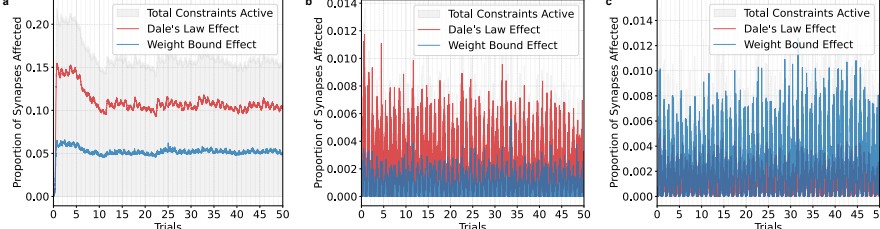

Figure S8: Analysis of constraint enforcement dynamics for the three selected rule sets from Fig. 6. The rules were evaluated over 50 training trials; the curves show the proportion of synapses triggering Dale's Law or hitting the maximum weight bound at each timestep. **a**. Overall best-performing rule (form: $\Delta w = X_{\text{post}} + S_{\text{pre}} \cdot X_{\text{pre}} + S_{\text{post}} \cdot X_{\text{pre}} + X_{\text{post}}^2 + X_{\text{post}} \cdot R$). **b**. Best rank 1 rule (form: $\Delta w = S_{\text{pre}}$). **c**. Best rank 2 rule. (form: $\Delta w = S_{\text{pre}} + S_{\text{pre}} \cdot X_{\text{post}}$)

## F    ADDITIONAL RESULTS OF PLASTICITY RULE

### F.1    CONSTRAINT ENFORCEMENT DYNAMICS

To analyze the functionality of the constraints, we tracked the enforcement dynamics for the three representative rules highlighted with color in Table 1 and mentioned in Fig. 6. We observed significant differences in how distinct plasticity rules interact with these boundaries. Fig. S8a shows that the best performing rule actively engages the constraints, with approximately 15% of synapses frequently hitting the hard bounds or Dale's law limits during training. In contrast, other rules (Fig. S8b and S8c) remain almost entirely within the permissible range. Importantly, our evolutionary optimization process does not penalize boundary contact; rather, we view these constraints as integral components of the synaptic dynamics. From an evolutionary perspective, the existence of these hard bounds serves as a stabilizing scaffold, allowing the search algorithm to explore more complex plasticity profiles that drive rapid learning. Without these bounds, such rules would very likely lead to catastrophic weight explosion or sign reversal, especially in the early stages when the parameters are not fully tuned.

### F.2    DISTRIBUTION OF DISCOVERED PLASTICITY RULES

As shown in Table S4, we identified and filtered 70 candidate plasticity rules from the Pareto-optimal population from the last generation based on overall task performance and biological validity criteria. The prevalent mathematical formulations were ranked and selected according to their ratio of occurrence within the filtered population. Fig. S9 illustrates the visualization of synapse-type-specific coefficients for representative plasticity rules from the filtered rule population in Table S4. Empty rows indicate that the corresponding term is absent from all rules in the filtered population. Each polar plot shows the coefficient distribution of one plasticity candidate term. Panels (a-d) display coefficient distributions for synapse-type-specific plasticity terms that depend on both presynaptic and postsynaptic neuron types (e.g., $S_{pre} \cdot S_{post}$ or $X_{pre} \cdot X_{post}$), organized by connection class: (a) excitatory-to-excitatory, (b) excitatory-to-inhibitory, (c) inhibitory-to-excitatory, and (d) inhibitory-to-inhibitory connections. Panel (e) shows coefficient distributions for neuron-type-dependent terms that depend only on either presynaptic or postsynaptic activity (e.g., $S_{pre}$ or $X_{post}$), which are shared across all types of connections.

### F.3    LAYER-WISE FIRING RATE DYNAMICS

Fig. S10 complements Fig. 5a by visualizing the layer-specific average firing rate dynamics during the test phase, optimized using the overall best-performing plasticity rule identified from the filtered rule population. The result demonstrates that the optimized V1 model effectively sustains persistent activity across all cortical layers during the delay period.

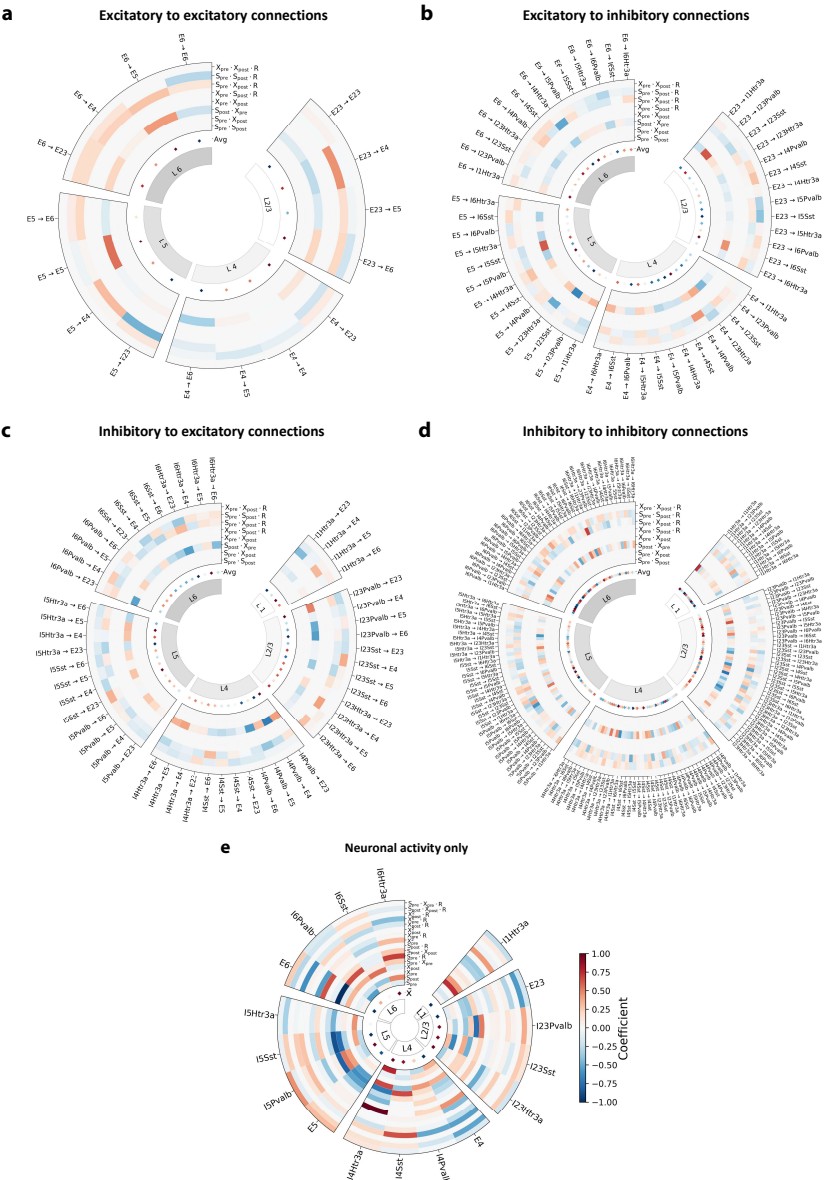

Figure S9: Illustration of how different rules from the filtered population as shown in Table S4 distribute their plasticity coefficients across the 25 plasticity candidate terms shown in Sec. D.1.

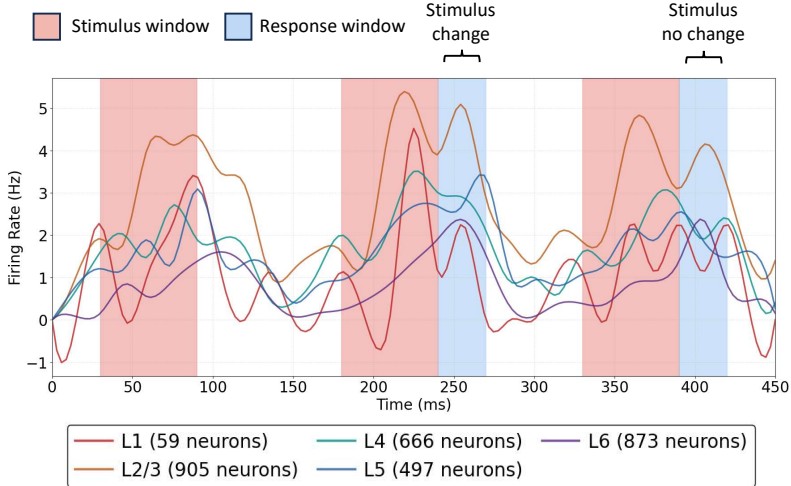

Figure S10: Visualization of cortical layer-specific firing rate dynamics during the test phase, optimized using the overall best-performing plasticity rule identified from the filtered rule population (form: $\Delta w = X_{\text{post}} + S_{\text{pre}} \cdot X_{\text{pre}} + S_{\text{post}} \cdot X_{\text{pre}} + X_{\text{post}}^2 + X_{\text{post}} \cdot R$).

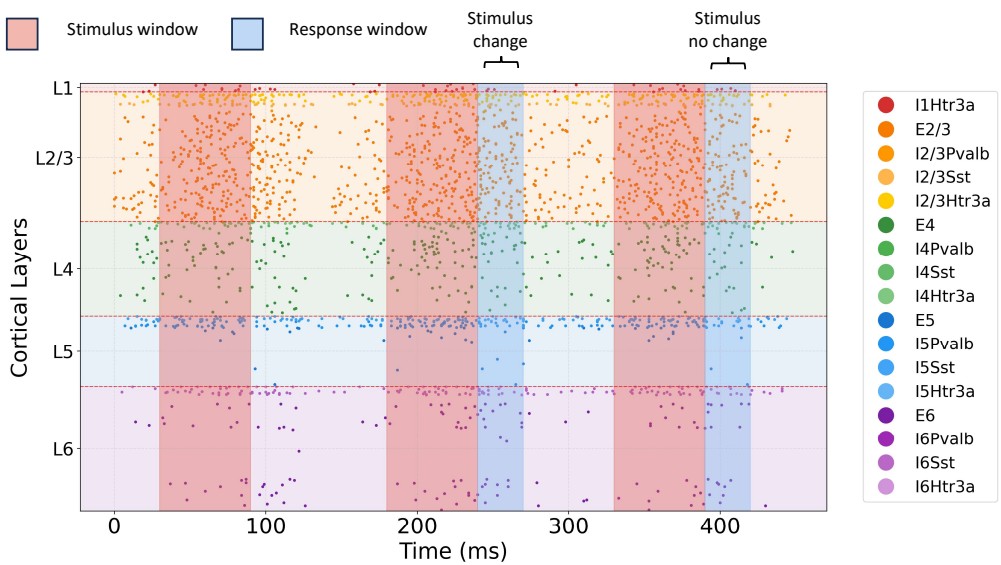

Figure S11: Spike raster plot during the test phase, optimized using a *reward-free* plasticity rule with high performance identified from the filtered rule population (form: $\Delta w = S_{pre}$).

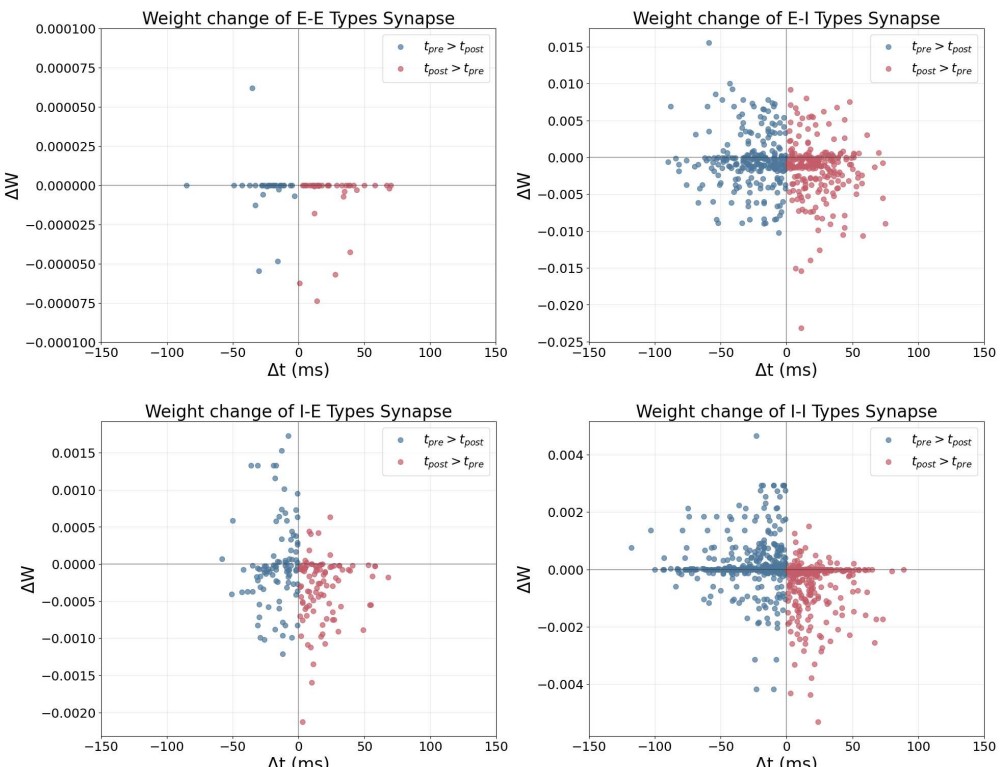

Figure S12: Visualization of the overall best-performing plasticity rule.

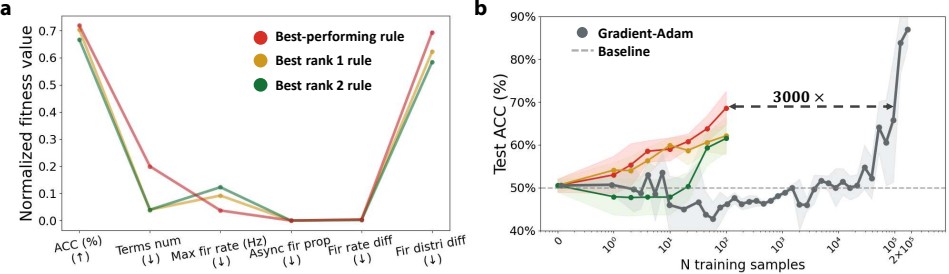

Figure S13: Comparison of performance on visual change detection of *natural images* between three representative rules and the widely used gradient-based Adam optimizer. **a**. Performance overview. **b**. Mean test accuracy over 5 seeds (200 test trials each); shaded areas indicate standard deviation.

| Rank | Mathematical formulation of plasticity rules in filtered population | Ratio | Obj-1 (↑) | | Obj-2 (↓) | | Obj-3 (↓) | | Obj-4 (↓) | | Obj-5 (↓) | | Obj-6 (↓) | |
|---|---|---|---|---|---|---|---|---|---|---|---|---|---|---|
| | | | Avg | Best | Avg | Best | Avg | Best | Avg | Best | Avg | Best | Avg | Best |
| 1 | $\Delta w = S_{\text{pre}}$ | 46.67% | 65.19 | 70.29 | 0.04 | 0.04 | 156.99 | 127.83 | 0.00 | 0.00 | 1.25 | 0.86 | 0.83 | 0.71 |
| 2 | $\Delta w = S_{\text{pre}} \cdot X_{\text{post}}$ | 10.67% | 63.91 | 68.43 | 0.04 | 0.04 | 165.88 | 125.96 | 0.00 | 0.00 | 1.14 | 0.87 | 0.92 | 0.86 |
| 3 | $\Delta w = X_{\text{pre}} \cdot R$ | 5.33% | 61.57 | 64.64 | 0.04 | 0.04 | 162.09 | 133.94 | 0.02 | 0.00 | 5.39 | 1.06 | 0.87 | 0.78 |
| 4 | $\Delta w = S_{\text{post}} \cdot X_{\text{pre}}$ | 5.33% | 62.50 | 63.21 | 0.04 | 0.04 | 170.62 | 137.43 | 0.00 | 0.00 | 1.04 | 0.84 | 0.91 | 0.85 |
| 5 | $\Delta w = S_{\text{post}}$ | 4.00% | 61.95 | 63.57 | 0.04 | 0.04 | 191.33 | 183.74 | 0.00 | 0.00 | 0.85 | 0.75 | 1.03 | 0.99 |
| 6 | $\Delta w = S_{\text{pre}} \cdot X_{\text{pre}}$ | 4.00% | 65.65 | 67.07 | 0.04 | 0.04 | 167.58 | 149.37 | 0.00 | 0.00 | 1.06 | 0.76 | 0.85 | 0.84 |
| 7 | $\Delta w = S_{\text{pre}} + S_{\text{pre}} \cdot X_{\text{post}}$ | 2.67% | 69.79 | 69.86 | 0.08 | 0.08 | 175.01 | 168.95 | 0.00 | 0.00 | 1.33 | 1.18 | 0.73 | 0.62 |
| 8 | $\Delta w = S_{\text{pre}} + S_{\text{pre}} \cdot X_{\text{pre}}$ | 2.67% | 65.89 | 67.36 | 0.08 | 0.08 | 137.04 | 133.72 | 0.00 | 0.00 | 1.36 | 1.20 | 0.88 | 0.86 |
| 9 | $\Delta w = S_{\text{post}} \cdot X_{\text{post}}$ | 2.67% | 62.36 | 62.79 | 0.04 | 0.04 | 190.78 | 187.54 | 0.00 | 0.00 | 0.97 | 0.73 | 1.05 | 1.05 |
| 10 | $\Delta w = X_{\text{post}} + S_{\text{pre}} \cdot X_{\text{pre}} + S_{\text{post}} \cdot X_{\text{pre}} + X_{\text{post}}^2 + X_{\text{post}} \cdot R$ | 1.33% | 71.86 | 71.86 | 0.20 | 0.20 | 115.63 | 115.63 | 0.00 | 0.00 | 1.76 | 1.76 | 0.98 | 0.98 |
| 11 | $\Delta w = S_{\text{pre}} + S_{\text{pre}} \cdot X_{\text{post}} + S_{\text{pre}} \cdot X_{\text{post}} \cdot R + S_{\text{post}} \cdot X_{\text{post}} \cdot R$ | 1.33% | 60.21 | 60.21 | 0.16 | 0.16 | 138.89 | 138.89 | 0.00 | 0.00 | 1.03 | 1.03 | 0.79 | 0.79 |
| 12 | $\Delta w = S_{\text{pre}} \cdot X_{\text{post}} + S_{\text{pre}} \cdot X_{\text{post}} \cdot R$ | 1.33% | 61.07 | 61.07 | 0.08 | 0.08 | 119.51 | 119.51 | 0.00 | 0.00 | 1.21 | 1.21 | 0.92 | 0.92 |
| 13 | $\Delta w = S_{\text{post}} + S_{\text{post}} \cdot X_{\text{post}} \cdot R$ | 1.33% | 66.57 | 66.57 | 0.08 | 0.08 | 130.64 | 130.64 | 0.00 | 0.00 | 1.36 | 1.36 | 0.90 | 0.90 |
| 14 | $\Delta w = S_{\text{pre}} + S_{\text{pre}} \cdot X_{\text{post}} + S_{\text{pre}} \cdot X_{\text{pre}} \cdot R$ | 1.33% | 65.93 | 65.93 | 0.12 | 0.12 | 126.65 | 126.65 | 0.00 | 0.00 | 1.33 | 1.33 | 0.78 | 0.78 |
| 15 | $\Delta w = X_{\text{post}}$ | 1.33% | 62.36 | 62.36 | 0.04 | 0.04 | 182.80 | 182.80 | 0.00 | 0.00 | 0.87 | 0.87 | 1.06 | 1.06 |
| 16 | $\Delta w = S_{\text{post}} \cdot R$ | 1.33% | 60.71 | 60.71 | 0.04 | 0.04 | 132.30 | 132.30 | 0.00 | 0.00 | 1.25 | 1.25 | 0.83 | 0.83 |
| 17 | $\Delta w = S_{\text{pre}} + S_{\text{pre}} \cdot X_{\text{post}} \cdot R$ | 1.33% | 66.29 | 66.29 | 0.08 | 0.08 | 162.30 | 162.30 | 0.00 | 0.00 | 0.97 | 0.97 | 0.80 | 0.80 |
| 18 | $\Delta w = S_{\text{pre}} \cdot X_{\text{post}} + S_{\text{post}} \cdot X_{\text{post}}$ | 1.33% | 63.58 | 63.58 | 0.08 | 0.08 | 131.29 | 131.29 | 0.00 | 0.00 | 1.35 | 1.35 | 0.94 | 0.94 |
| 19 | $\Delta w = S_{\text{pre}} + X_{\text{pre}}^2 \cdot R$ | 1.33% | 67.29 | 67.29 | 0.08 | 0.08 | 150.14 | 150.14 | 0.00 | 0.00 | 1.15 | 1.15 | 0.85 | 0.85 |
| 20 | $\Delta w = S_{\text{pre}} + S_{\text{post}} \cdot X_{\text{post}} \cdot R$ | 1.33% | 66.21 | 66.21 | 0.08 | 0.08 | 156.19 | 156.19 | 0.00 | 0.00 | 1.09 | 1.09 | 0.82 | 0.82 |
| 21 | $\Delta w = S_{pre} + S_{pre} \cdot X_{pre} + S_{pre} \cdot X_{pre} \cdot R$ | 1.33% | 67.43 | 67.43 | 0.12 | 0.12 | 155.35 | 155.35 | 0.00 | 0.00 | 1.11 | 1.11 | 0.84 | 0.84 |

Table S4: Mathematical formulations of all plasticity rules ranked by ratio of occurrence in the *filtered Pareto-optimal population* and their average performance on six objectives. Each formulation may contain multiple rules with identical mathematical form but varying hyperparameters. *Avg*: average fitness across individuals within each formulation; *Best*: optimal fitness within each formulation.

## F.4 REWARD-FREE PLASTICITY RULE DYNAMICS

Fig. S11 also complements Fig. 5a by visualizing the spike raster plot of V1 model optimized using a *reward-free* plasticity rule with high performance identified from the filtered rule population. The result demonstrates that even without reward modulation, the optimized V1 model can still maintain stable firing dynamics during the test phase. Especially, different types of neurons exhibit diverse firing patterns during the stimulus presentation, delay periods and response windows.

## F.5 SYNAPTIC WEIGHT CHANGES OF THE OVERALL BEST-PERFORMING RULE

Fig. S12 illustrates the synaptic weight changes induced by the discovered overall best-performing plasticity rule, $\Delta w = X_{post} + S_{pre} \cdot X_{pre} + S_{post} \cdot X_{pre} + X_{post}^2 + X_{post} \cdot R$, across coarser-grain categories of synaptic connections: excitatory-to-excitatory, excitatory-to-inhibitory, inhibitory-to-excitatory, and inhibitory-to-inhibitory.

## F.6 COMPARISON WITH GRADIENT-BASED OPTIMIZATION ON NATURAL IMAGES

Fig. S13 illustrates the performance comparison between three representative plasticity rules as defined in Sec. 3.3 in the main manuscript and the gradient-based Adam optimizer on a visual change detection task with natural images. For clarity, we briefly restate the definitions of the three selected rules: (i) the overall best-performing rule with formulation $\Delta w = X_{post} + S_{pre} \cdot X_{pre} + S_{post} \cdot X_{pre} + X_{post}^2 + X_{post} \cdot R$, (ii) the best rank-1 rule, defined as the optimal performer within the $\Delta w = S_{pre}$ class, and (iii) the best rank-2 rule, representing the top performer in the $\Delta w = S_{pre} \cdot X_{post}$ category.

## F.7 LONG TIME HORIZON TRAINING AND HOMEOSTATIC PROPERTY

To further characterize the stability properties of the discovered plasticity mechanisms, we evaluated the overall best-performing rule, $\Delta w = X_{post} + S_{pre} \cdot X_{pre} + S_{post} \cdot X_{pre} + X_{post}^2 + X_{post} \cdot R$, under extended training regimes beyond its evolutionary setting. As shown in Fig. S15, although the overall best-performing rule was evolved under a 100 training trials' setting, it exhibits consistent performance stabilization when exposed to orders of more training samples. This sustained adaptability suggests that the discovered plasticity mechanism possesses intrinsic self-regulatory dynamics that prevent performance degradation during continued fine-tuning, analogous to homeostatic plasticity observed in biological neural circuits.

However, we need to acknowledge that this robustness is non-trivial and not representative of the entire filtered explored rule population. The majority of evolved rules exhibit performance degradation under prolonged exposure to stimuli. This fragility is especially evident in reward-free plasticity rules, where the lack of modulatory feedback renders the network prone to unbounded synaptic drift and functional collapse outside the evolutionary setting.

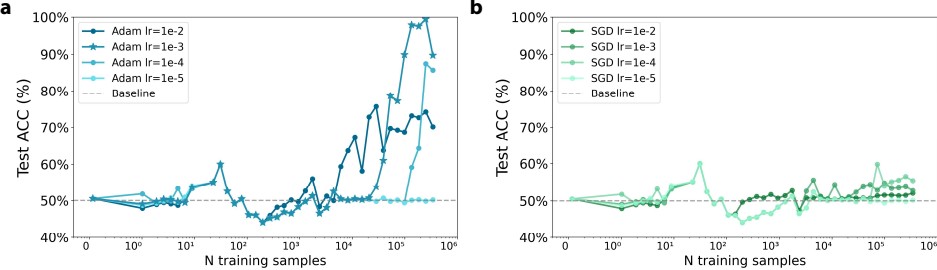

Figure S14: Sensitivity analysis training V1 model on visual change detection on natural images via Adam and SGD with different *learning rates*. The best performance is achieved using Adam with a learning rate of $10^{-3}$, which is highlighted with $\star$ marker. **a**. Adam optimizer. **b**. SGD optimizer.

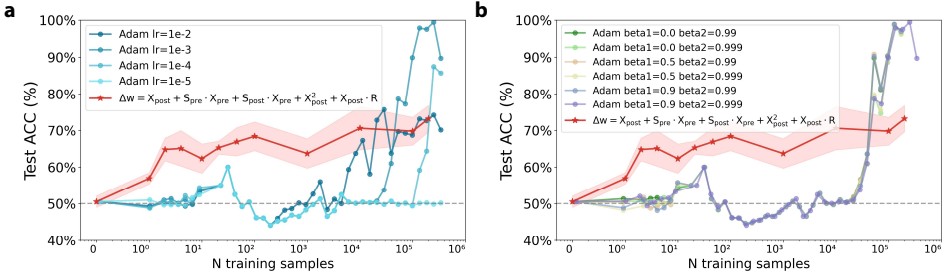

Figure S15: **a**. Task performance dynamics of the explored overall best-performing plasticity rule with sufficient training samples on visual change detection on natural images. **b**. Sensitivity analysis of the Adam optimizer's performance on visual change detection on natural images with respect to the $\beta_1$ and $\beta_2$ *momentum hyperparameters*. The learning rate is fixed at $10^{-3}$. Each data point shows the test accuracy averaged over its 50 nearest neighbors to mitigate sampling variability. The overall best-performing synaptic plasticity rule is also provided for reference, marked with $\star$.

## G   GRADIENT DESCENT SETTINGS AND SENSITIVITY ANALYSIS

### G.1   DETAILED GRADIENT BASED OPTIMIZATION SETTINGS

For the gradient-based results in the main manuscript, networks were trained using the Adam optimizer with a learning rate of 0.001 and default values for all other hyperparameters. This setting was chosen based on a sensitivity analysis in Sec. G.2.

To enable gradient flow through spiking neurons, a Gaussian surrogate gradient was implemented following Chen et al. (2022). The surrogate gradient was parameterized with a fixed standard deviation $\sigma = 0.5$ and amplitude $= 0.3$, applied to the scaled membrane potential. These parameters were not updated during training. Backpropagation through time (BPTT) was used for credit assignment across temporal sequences. This setup ensures a direct comparison between gradient-based optimization and the explored plasticity rules, while accounting for the additional computational cost introduced by surrogate gradients and BPTT.

### G.2   SENSITIVITY ANALYSIS IN HYPERPARAMETERS OF OPTIMIZERS

In Fig. S14a, we present the results of training the V1 model on the visual change detection task with natural images using the Adam optimizer across a range of learning rates: $\{10^{-5}, 10^{-4}, 10^{-3}, 10^{-2}\}$. The analysis reveals that the V1 model is highly sensitive to optimizer choice, where the model achieves peak performance with a learning rate of $10^{-3}$, while other rates degrade convergence. We also evaluated SGD with the same set of learning rates for comparison in Fig. S14b. However, SGD fails to converge effectively across all tested learning rates within the same training budget.

We extended our analysis to the sensitivity of Adam's momentum hyperparameters, $\beta_1$ and $\beta_2$, with the learning rate fixed at previously identified optimal value, $10^{-3}$. As shown in Fig. S15b, performance remains stable across $\beta_1 \in \{0, 0.5, 0.9\}$ and $\beta_2 \in \{0.99, 0.999\}$. This relative insensitivity to $\beta$ parameters reinforces our conclusion that evolved plasticity rules achieve behavioral performance with order-of-magnitude greater efficiency than gradient-based methods.

## H  Details of multi-objective evolutionary framework

As outlined in Section 2, the discovery of plasticity mechanisms can be cast as a multi-objective search problem. Each candidate corresponds to a parameterized plasticity rule, and the goal is to identify rules that strike a balance between predictive performance, biological plausibility, and computational efficiency. This search problem presents several key challenges:

**Expensive evaluation**. Evaluating the quality of a plasticity rule requires training a new V1 network from scratch under the candidate rule. This involves long simulation times and limits the number of rules that can be explored within practical runtime.

**Many objectives**. The optimization is multi-objective, combining quantitative performance (*e.g.*, task accuracy, number of terms) with qualitative desiderata. Some objectives are *soft-bounded*: for instance, it is desirable for the learned plasticity rule to exhibit dynamics similar to biological systems, but perfect matching is neither required nor expected. This leaves room for approximate but functionally relevant behaviors.

**High-dimensional search space**. The plasticity rule is defined over multiple interacting components. This leads to a large combinatorial search space with complex dependencies.

**Noisy evaluation**. Because a plasticity rule must be assessed by training a network from scratch on varying data, its measured performance inevitably exhibits a substantial degree of randomness. As a result, evaluations of the same rule can differ across runs. An effective search framework must therefore be robust to this variability, avoiding misleading selections and favoring rules that demonstrate consistently strong performance.

**Non-differentiable optimization**. The objective functions do not admit closed-form gradients with respect to the rule parameters. This rules out traditional gradient-based optimization and motivates the use of derivative-free search.

Taken together, these factors make the search for plasticity mechanisms especially demanding. To address them, we employ a evolutionary framework, which is well suited to handling non-differentiability, noise, and multiple objectives while naturally supporting parallel and scalable evaluation. The following section details the specific evolutionary algorithm used in our study.

### H.1  Proposed Evolutionary Framework

To address the challenges outlined above, we propose a tailored evolutionary framework for plasticity rule discovery. Our method builds on established multi-objective evolutionary algorithms (MOEAs) and is implemented within the EvoX framework (Huang et al., 2024a).

The algorithm is designed to be robust to noise, scalable to large populations, and efficient on modern GPU clusters through parallel evaluation. Its workflow follows four main phases: *initialization*, *reproduction*, *evaluation* (including re-evaluation), and *selection*, as summarized in Algorithm 1.

During *initialization*, a population of candidate rules is randomly sampled from the parameter space. In each generation, *reproduction* produces offspring via swarm-inspired updates followed by mutation. *Evaluation* then proceeds on the offspring and on a dynamically selected subset of the current population for re-evaluation (i.e., individuals with few prior evaluations). This adaptive re-evaluation saves computational budget early while preserving the ability to reliably discriminate between similarly performing solutions in later stages through multiple rounds of evaluation. Finally, *selection* ranks all evaluated solutions using a noise-aware non-dominated sorting procedure and retains the most promising candidates for the next generation.

#### H.1.1  Reproduction Operator

The reproduction operator follows a Competitive Swarm Optimizer (CSO)–style mechanism adapted from TensorRVEA (Cheng & Jin, 2015; Liang et al., 2024). The population is randomly partitioned into $N/2$ disjoint pairs. For each pair, a *teacher* (winner) and a *student* (loser) are identified by comparing their performance on a randomly sampled objective, where the objective index is drawn according to a predefined weight distribution. *Only the student* updates its velocity and position by combining directional cues from the teacher and the population center; the teacher re-

---

**Algorithm 1** Noise-Aware Multi-Objective Evolutionary Framework

---

1: Initialize population $P$ with random candidates
2: Initialize evaluation statistics (counts, means, variances)
3: **while** termination criteria not met **do**
4:      Offspring $\leftarrow$ REPRODUCTION$(P)$                ▷ Sec. H.1.1
5:      $\mathcal{R} \leftarrow$ select subset of $P$ with lowest evaluation counts for re-evaluation
6:      Evaluate Offspring $\cup \, \mathcal{R}$ in parallel         ▷ Sec. H.1.3
7:      Update evaluation statistics
8:      $P \leftarrow$ SELECTION$(P, \text{Offspring}, \mathcal{R})$          ▷ Sec. H.1.2
9: **end while**
10: **return** Final non-dominated set from $P$

---

**Algorithm 2** Reproduction$(P, \{\mathbf{v}_i\}, \phi)$

---

1: **Input:** Population $P = \{\mathbf{x}_i\}_{i=1}^N$, velocities $\{\mathbf{v}_i\}_{i=1}^N$, convergence parameter $\phi$, objective weights $\mathbf{w} \in \mathbb{R}^m$
2: **Output:** Offspring set $P' = \{\mathbf{x}_i'\}_{i=1}^N$ and offspring velocities $\{\mathbf{v}_i'\}_{i=1}^N$
3: Randomly shuffle indices and form $N/2$ disjoint pairs
4: Compute population center $\mathbf{c} \leftarrow \frac{1}{N} \sum_{i=1}^N \mathbf{x}_i$            (parallel reduction)
5: **for** each pair $(i, j)$ **in parallel do**
6:      Sample objective $k \sim$ Categorical$(\mathbf{w})$
7:      Determine teacher/student by objective-wise comparison (assume minimization):

$$\text{if } f_k(\mathbf{x}_i) \leq f_k(\mathbf{x}_j): \ t \leftarrow i, \ s \leftarrow j \ \text{ else } \ t \leftarrow j, \ s \leftarrow i$$

8:      Sample elementwise coefficients $\boldsymbol{\lambda}_1, \boldsymbol{\lambda}_2, \boldsymbol{\lambda}_3 \sim \mathcal{U}[0,1]^d$
9:      **Student update (only):**

$$\mathbf{v}_s \leftarrow \boldsymbol{\lambda}_1 \odot \mathbf{v}_s + \boldsymbol{\lambda}_2 \odot (\mathbf{x}_t - \mathbf{x}_s) + \boldsymbol{\lambda}_3 \odot \phi \odot (\mathbf{c} - \mathbf{x}_s)$$

$$\tilde{\mathbf{x}}_s \leftarrow \mathbf{x}_s + \mathbf{v}_s$$

10:      **Mutation (both members produce offspring):**
       $\mathbf{x}_s' \leftarrow$ MUTATE$(\tilde{\mathbf{x}}_s)$    (continuous: polynomial; gating: discrete/bit-flip; dimension-aware rates)
       $\mathbf{x}_t' \leftarrow$ MUTATE$(\mathbf{x}_t)$                (teacher does not move before mutation)
11:      Set offspring velocities: $\mathbf{v}_s' \leftarrow \mathbf{v}_s$ (updated), $\mathbf{v}_t' \leftarrow \mathbf{v}_t$ (unchanged)
12:      Add $\mathbf{x}_s', \mathbf{x}_t'$ (and $\mathbf{v}_s', \mathbf{v}_t'$) to $P'$
13: **end for**
14: **return** $P', \{\mathbf{v}_i'\}_{i=1}^N$

---

mains stationary in this phase. To produce a full set of $N$ offspring per generation, *both* members of the pair then undergo mutation: one offspring is generated from the updated student, and a second offspring is generated from the teacher.

This CSO-style pairing and update are well suited to GPU acceleration: pair formation, pair-wise comparison, per-dimension velocity updates, and independent mutations can be executed in parallel without data dependencies (the population center is a single reduction).

### H.1.2 SELECTION OPERATOR

The selection phase accounts for noisy evaluations by incorporating statistical confidence into the dominance relation used in non-dominated sorting (Deb et al., 2002) while following the general principle of dealing with noisy data in evolutionary algorithm (Jin & Branke, 2005; Fieldsend & Everson, 2005). Each solution maintains online estimates of the mean and variance of its objectives, updated via Welford's method. Rather than applying deterministic dominance, solutions are compared probabilistically, the algorithm estimates the probability that one solution outperforms another using their means and variances.

Formally, for each objective, a win-rate probability is computed between two solutions. A solution is said to dominate another only if it achieves a sufficiently high probability (*e.g.*, $> 55\%$) of being better on *all* objectives. This definition naturally adapts to noise: under high variance, dominance requires a clear margin of superiority, while under low variance, even small improvements can suffice. In this way, the selection pressure adjusts automatically to evaluation reliability.

These probabilistic dominance relations are used to construct a dominance matrix, which forms the basis for a confidence-aware non-dominated sorting procedure. After sorting, ties near the population cutoff are broken using a diversity measure (crowding distance) adjusted with an uncertainty penalty, favoring solutions that are both diverse and reliably evaluated.

This procedure reduces bias from noisy evaluations, improving stability in stochastic environments and enabling reliable optimization when evaluations are expensive and uncertain. The overall process is summarized in Algorithm 3.

---

**Algorithm 3** Selection with Probabilistic Non-Dominated Sorting

---

1: **Input:** Candidate set $C$ (parents, offspring, re-evaluated), mean objectives $\bar{\mathbf{f}}$, variances $\mathbf{s}^2$, population size $N$
2: **Output:** Survivor set $P$
3: Update $\bar{\mathbf{f}}$ and $\mathbf{s}^2$ for all $x \in C$ via online estimators
4: **for** each pair $(i, j)$ in $C$ **do**
5:     **for** each objective $k$ **do**
6:         Compute win probability:

$$\text{WinRate}_{ij}^{(k)} = \frac{1}{1 + \exp\left(-\frac{\bar{f}_j^{(k)} - \bar{f}_i^{(k)}}{\sqrt{s_i^{2(k)} + s_j^{2(k)} + \epsilon}}\right)}$$

7:     **end for**
8:     $i$ dominates $j$ if $\text{WinRate}_{ij}^{(k)} > \tau$ for all $k$
9: **end for**
10: Build dominance matrix and perform non-dominated sorting
11: Break ties at cutoff using adjusted crowding distance:

$$\text{AdjDist}_i = \text{Crowding}_i - \kappa \cdot \frac{1}{m} \sum_{k=1}^{m} \frac{s_i^{2(k)}}{\max_j s_j^{2(k)}}$$

12: Select top $N$ individuals as survivor set $P$

---

### H.1.3 Efficient Evaluation

Evaluating a candidate plasticity rule is computationally demanding, as each evaluation requires training a V1 model from scratch. To make this feasible at scale, two key complementary strategies are explored: batched parallelism and an optimized forward pass enabled by the gradient-free setup. Together, these techniques substantially reduce computational and memory costs, making large-scale exploration of candidate rules tractable.

**Batched evaluation**. At each generation, $P$ candidate rules are trained concurrently. State variables (membrane potentials, spikes, synaptic traces) are organized in a Structure-of-Arrays (SoA) layout, rather than an Array-of-Structures (AoS). This design improves coalesced memory access and GPU bandwidth utilization, allowing all candidate rules to advance in lockstep. Within a single device, data parallelism enables simultaneous processing of multiple candidates and mini-batches. The same principle extends across multiple devices or nodes, achieving near-linear scaling as computational resources increase.

**Optimized forward pass (Gradient-free setup)**. Because the evolutionary search is gradient-free, no backpropagation-through-time is required (Werbos, 1990). This removes the need to store long sequences of activations, so memory usage depends only on the instantaneous network state rather than the simulation length. As a result, models can be trained over long horizons by the plastic-

ity rules without the prohibitive costs of differentiable approaches. The gradient-free setup also aligns naturally with the event-driven nature of spiking models. In contrast to surrogate-gradient methods (Huh & Sejnowski, 2018; Zenke & Ganguli, 2018; Neftci et al., 2019b), spikes are treated directly as binary events, which can be stored and processed compactly. This representation reduces both memory bandwidth demands and computation relative to floating-point operations. When combined with batched evaluation, these properties make forward-only simulation particularly efficient.

These efficiencies stem directly from the design of an evolutionary framework. Its gradient-free nature enables many techniques that avoid the heavy compute and memory demands of backpropagation, while its population-based structure inherently supports batching and large-scale parallelism. Taken together, these properties allow our system to sustain evaluation throughput on the order of $10^5$ candidate plasticity rules per GPU per day, with each rule applied to training a V1 model of 3,000 neurons and approximately 400k parameters from scratch on two distinct tasks. This level of scalability enables exploration of vast rule spaces that would otherwise be computationally prohibitive.

### Large Language Model usage disclosure

In accordance with the policy on LLM usage, we provide the following disclosure regarding how LLMs were employed during this work. LLMs were used only in supportive roles, with all substantive research contributions, coding, and writing decisions made by the human authors.

**Code base**. LLMs were used strictly for short code auto-completion, comparable to standard IDE assistance. All completions were reviewed and confirmed by the human authors. No unsupervised code generation, automatic project generation, or independent algorithm design was delegated to an LLM.

**Paper writing**. LLMs assisted only at the presentation level, *e.g.*, suggesting alternative phrasings or improving clarity of exposition. All scientific content, descriptions of methodology, results, and discussion were written and verified by the authors. All language suggestions from the LLM were reviewed by the authors.

**Research assistant**. LLMs were occasionally used as research assistants in a role analogous to a search engine. This included tasks such as quickly gathering related information or checking terminology. All final written content is based on references and verified sources; no scientific claims or results were drawn solely from LLM outputs.

