# OpenReview forum: "Discovering heterogeneous synaptic plasticity rules via large-scale neural evolution"
_ICLR.cc/2026/Conference — ICLR 2026 Poster_

### Official Review · Reviewer_t465 · 2025-10-28

**Soundness:** 3
**Presentation:** 2
**Contribution:** 2
**Rating:** 6
**Confidence:** 3

**Summary:**

The paper presents a computational framework that uses Darwinian evolution to automatically discover biologically plausible, heterogeneous synaptic plasticity rules in a realistic model of mouse primary visual cortex (V1). By parameterizing plasticity as a Taylor expansion of pre- and postsynaptic spikes, eligibility traces, and neuromodulators, the authors create a compact space plasticity rules. A multi-objective evolutionary algorithm optimizes rules for both task performance and biological validity. The resulting rules achieve high performance on visual change-detection tasks with far fewer samples than gradient-based methods, generalize to networks of different sizes, and maintain biological constraints, offering insights into how the brain might implement efficient, heterogeneous plasticity.

**Strengths:**

- First large-scale attempt to let evolution simultaneously optimize thousands of plasticity coefficients inside a biological cortical model.
Introduces a compact closed-form plasticity rules that keeps rules local and therefore biologically testable (Up to 25 terms, far surpassing existing plasticity rules containing 2 to 4 terms).
- Multi-objective fitness to control both the task performance and biological plausibility.
- If I understand it correctly, the exact same rules work after rescaling on networks from 1 k to 5k neurons without re-evolution.

**Weaknesses:**

- The work is an impressive proof-of-concept that evolutionary search can find local plasticity rules which learn quickly and remain biologically plausible, but it stops short of asking whether those rules generalize to new and practical tasks. My main concern after reading this paper is “what should the readers take away?” Sample efficiency of Hebbian plasticity has already been proved by many previous works. According to my knowledge, the real problem for Hebbian plasticity is “are there general plasticity rules that generalize any tasks & networks?” So far, the generalization of those local learning rules can not compare with that of gradient descent.
- The paper contains no statistical comparison with other biologically plausible learning rules.

**Questions:**

- The paper should explain the Evaluation Tasks (visual change detections) more clearly at least in the appendices, which is less clear for readers who are not familiar enough

- Can you explain where did the $c^k$ in equation 7 go in Table 1? Are Table 1 cluster different plasticity rules by $g^k$ only? I think $c^k$ is real number and is non-homogeneous for different synapses. Why is it possible to collapse to 4000 unique rules?

- Did the paper study the problem of generalization? What is the relation between training/testing and training/validation trials? Are those tasks unchanged? Or are the tasks different? How different they are?

- It seems to me that evolution are done on a fixed V1 model, while the evolved plasticity rules can be applied to train different networks / networks of different size? Please clarify more on the experiment settings.

---

> ### Author Response · Authors · 2025-11-22
> **Response to Reviewer t465 [1/3]**
>
> Thanks for your careful review and comments!
>
> We value your assessment that our work offers insights into how the brain might implement efficient heterogeneous plasticity.
>
> Please kindly find the point-to-point response to the questions and suggestions.
>
> ## Major Suggestions and Questions:
> 1. What should the readers take away from this paper, as the sample efficiency of Hebbian plasticity has already been proved by many previous works.
>
>    - We thank the reviewer for raising this important question about our paper's take-home message.
>
>    - We have added a **Discussion** section and revised the **Abstract**, **Introduction** and **Conclusion** sections to provide clearer distinction between this work and previous studies, and key takeaway.
>
>    - In short, the key contributions of this work are:
>
>      - **A task-agnostic framework for plasticity rule exploration**: We introduce a principled framework that discovers plasticity rules satisfying both task performance and biological plausibility constraints. Unlike prior work limited to homogeneous rules or E/I heterogeneity, our approach explores synapse-type-specific mechanisms in an experimentally grounded V1 model [1].
>
>      - **Diverse plasticity rules show functional equivalence on fixed-task settings**: In revised Sec. 4.1, we discovered that multiple plasticity rules with markedly different mathematical forms can achieve comparable functional outcomes for task settings with stable environmental structure (like the visual change detection paradigm in this work). This degeneracy suggests that biological systems may leverage diverse synaptic mechanisms to achieve similar computational goals, enhancing robustness and adaptability.
>
>      - **Pre-synaptic-only plasticity mechanisms and their biological relevance in memory**: We discovered that purely pre-synaptic-dependent rules (e.g., $Δw = S_{pre}$) can solve the reward-required task without post-synaptic activity dependence.
>
>         - In the revised Sec. 4.2, we demonstrate that this finding shows strong correlation with experimental observations: environmental and state changes reshape `presynaptic` Ca²⁺ distributions under `minisecond-level time scale` [2] and receptor expression [3], thereby modulating synaptic strength without postsynaptic information` involve.
>
>         - This finding also complementary to Behavior timescale synaptic plasticity (BTSP) mechanisms relying on pre-synaptic activity which has been experimentally observed in hippocampus and works at `second-level time scale` [4].
>
>        - Furthermore, our discovery that pre-synaptic-only rules enable `working memory` capabilities. This finding shows correlation with recent work [5] which demonstrates that presynaptic activity alone can encode `episodic memory`.
>
>      - **Plasticity rules as carriers of innate abilities**: In our revised Sec. 4.3, we propose a plasticity-based perspective on `innate abilities`. Learning may involve not only interactions between plasticity rules and input stimuli, but also `evolutionary inductive biases` embedded within the plasticity rule structure and parameters. This few-shot learning analysis provided in this work potentially explains why many animals exhibit innate behavioral capabilities shortly after birth (e.g., odor-triggered suckling reflexes), as evolution has pre-configured appropriate learning mechanisms through ancestral selection pressures.

---

> > ### Author Response · Authors · 2025-11-22
> > **Response to Reviewer t465 [2/3]**
> >
> > 2. a) Whether the discovered plasticity rules can generalize to other tasks or networks? Did the paper study the problem of generalization? b) What is the relation between training/testing and training/validation trials? c) Are those tasks unchanged? Or are the tasks different? How different are they?
> >
> >    - a) Regarding generalization:
> >      - Currently, the visual change detection task is uniquely suited for our framework because it provides experimentally measured neuronal scale fine-grained activity of mice performing the same task [6,7]. These biological data enable our multi-objective optimization to constrain evolved rules by both functional performance and biological plausibility. This dual constraint is what distinguishes our work from prior plasticity rule discovery efforts.
> >      - The generalization analysis is given in Fig. 7, where the discovered plasticity rules are evaluated on larger or smaller V1 models with different random seeds. The results demonstrate that the evolved rules generalize well to networks of varying sizes and initializations within the same task context.
> >      - Incorporating additional tasks should indeed strengthen the generality of our conclusions. However, the V1 model is dedicated to visual processing, which makes switching to a non-visual task (like auditory) non-trivial without rebuilding the biological constraints. To the best of our knowledge, no other publicly available visual datasets provide comparable neuronal activity data for other tasks.
> >      - The proposed framework itself is task-agnostic. It is ready to be applied to other tasks once appropriate experimental constraints become available.
> >
> >     - b) Differences between training/testing and training/validation trials:
> >       - In our experiments, training trials are used during the evolutionary search to evaluate and select plasticity rules based on their performance and biological plausibility. Validation trials are held out and used only after the evolution process to assess the generalization of the discovered rules.
> >       - When the exploration of plasticity rules is complete, we evaluate the best-evolved rules on separate training/testing trials that were regenerated and not seen during the evolution. This allows us to measure how well the rules perform on entirely new data.
> >       - We have revised and provided a clearer definition in Sec.
> >
> >      - c) Task consistency:
> >        - The visual change detection task structure remains consistent throughout our experiments.
> >
> > 3. So far, the generalization of those local learning rules can not compare with that of gradient descent (GD).
> >
> >    - Yes. We agree that GD methods generally exhibit stronger generalization capabilities across diverse tasks and network architectures compared to local plasticity rules. Here is a brief clarification of the motivation for including the GD comparison. We have also provided the details in the revised Sec. 3.3, Sec. 4.3 to avoid potential misunderstanding.
> >
> >       - The GD comparison is NOT intended to show `general superiority` of explored plasticity rules over BP for algorithmic applications. Rather, we try to show the neuroscientific value of our framework via answering a scientific question: *Why do biological systems (e.g., humans) equipped with synaptic plasticity rules exhibit few-shot learning capabilities that differ qualitatively from artificial networks trained via GD?*
> >       - Our results reveal an important asymmetry: while gradient-based methods require *thousands of trials* and *mini-batch training* for stable optimization in the bio-data based V1 model, evolved plasticity rules under the same optimization constraints achieve comparable performance within ~100 samples. We speculate that this learning efficiency may stem from evolutionary embedding of task-related priors (i.e., *inductive bias* in AI) into the plasticity rule structure and parameters, since *the task setting is fixed*. Thus, the explored evolved plasticity mechanisms may encode ancestrally relevant computational strategies, enabling rapid adaptation without extensive trial-and-error learning.
> >       - This evolution process echoes to all species that evolve and interact with the environment, since the natural environment is also fixed. We call it a `plasticity` view to explain where the `innate ability` comes from. This distinguishes our work from neuromorphic computing efforts that propose plasticity rules as alternatives to backpropagation.

---

> > > ### Author Response · Authors · 2025-11-22
> > > **Response to Reviewer t465 [3/3]**
> > >
> > > 4. The paper should explain the Evaluation Tasks (visual change detections) more clearly, at least in the appendices, which are less clear for readers who are not familiar enough.
> > >
> > >    - We appreciate the reviewer's suggestion to clarify the evaluation task description. In response, we have expanded the explanation of the visual change detection task in Appendix E.2 of the revised manuscript. This section now provides a detailed overview of the task structure, stimulus presentation. We believe this addition will help readers unfamiliar with the task better understand its significance and implementation in our study.
> > >
> > > 3. a) Can you explain where did the $c^k$ in equation 7 go in Table 1? Are Table 1 cluster different plasticity rules by $g^k$ only? I think $c^k$ is real number and is non-homogeneous for different synapses. b) Why is it possible to collapse to 4000 unique rules?
> > >
> > >     - a) Regarding the grouping in Table 1: You are correct, the coefficients $c^k$ are real numbers and are non-homogeneous across different synapse types. In Table 1, we clustered the rules based solely on their mathematical structure (determined by the binary gates $g^k$) to highlight the functional forms.
> > >
> > >     - b) Regarding the 4,000 unique rules: The number 4,000 refers to the final population size of our evolutionary algorithm, not the number of unique mathematical possibilities. While the theoretical search space is infinite (due to continuous $c^k$), we maintain a fixed population of candidate rules per generation. This population allows us to sample the diversity of the landscape effectively while keeping the optimization tractable.
> > >
> > >
> > > 6. It seems to me that evolution is done on a fixed V1 model, while the evolved plasticity rules can be applied to train different networks / networks of different size? Please clarify more on the experiment settings.
> > >
> > >    - We want to clarify that while evolution occurs on a bio-data based V1 model, this is not a "fixed" model in the sense of fixed weights. The synaptic weights are dynamically modulated by the evolved plasticity rules in response to the task.
> > >    - In the revised **Discussion** section, we explore the connections between the plasticity rules explored by the proposed framework and experimentally observed plasticity mechanisms in biological systems, highlighting the explored plasticity rules' potential biological relevance and universality.
> > >    - Regarding generalization testing, we explicitly tested these evolved rules on V1 models with different scales and random initializations (see Fig. 7). The results demonstrate that the same plasticity rules produce consistent functional properties across different network instances, validating that they encode robust computational principles rather than overfitting to a specific configuration.
> > >
> > > ---
> > > Reference
> > >
> > > [1] Billeh, Y. N., Cai, B., Gratiy, S. L., Dai, K., Iyer, R., Gouwens, N. W., ... & Arkhipov, A. (2020). Systematic integration of structural and functional data into multi-scale models of mouse primary visual cortex. Neuron, 106(3), 388-403.
> > >
> > > [2] Cohn, R., Morantte, I., & Ruta, V. (2015). Coordinated and compartmentalized neuromodulation shapes sensory processing in Drosophila. Cell, 163(7), 1742-1755.
> > >
> > > [3] Handler, A., Graham, T. G., Cohn, R., Morantte, I., Siliciano, A. F., Zeng, J., ... & Ruta, V. (2019). Distinct dopamine receptor pathways underlie the temporal sensitivity of associative learning. Cell, 178(1), 60-75.
> > >
> > > [4] Bittner, K. C., Milstein, A. D., Grienberger, C., Romani, S., & Magee, J. C. (2017). Behavioral time scale synaptic plasticity underlies CA1 place fields. Science, 357(6355), 1033-1036.
> > >
> > > [5] Pang, R., & Recanatesi, S. (2025). A non-Hebbian code for episodic memory. Science Advances, 11(8), eado4112.

---

### Official Review · Reviewer_Txin · 2025-10-31

**Soundness:** 3
**Presentation:** 3
**Contribution:** 3
**Rating:** 8
**Confidence:** 3

**Summary:**

This paper proposes a synaptic plasticity evolutionary method that achieves a balance between task performance and biological interpretability through heterogeneous plasticity. This method is trained by finding an interpretable search space. Also, the plasticity algorithm they discovered has better generalization and the ability to learn from few shot learning.

**Strengths:**

**1.** It's a very interesting job with intriguing ideas. The ideas presented offer fresh perspectives and will likely stimulate further academic discussion.

**2.** The thesis is well-written, logical, and clear. The manuscript demonstrates good academic writing standards.

**3.** The research question is clearly defined.

**Weaknesses:**

**1.** The article is not very clear on how it assesses biological interpretability. Is it through the values related to firing rate in the metrics? Why would similar firing rates be more biologically similar? The methodology for evaluating biological interpretability requires clarification.

For example, How model units are aligned or matched with biological neurons, How the neural recordings from LGN were preprocessed?

**2.** There is no paragraph explaining how the similarity with the LGN regions is calculated and how the neural data is utilized.

**3.** Many formulas are presented without explaining the meanings of all the symbols. It is recommended to provide a complete explanation.

**Questions:**

See Weakness Section.

---

> ### Author Response · Authors · 2025-11-22
> **Response to Reviewer Txin**
>
> We sincerely appreciate the reviewer's valuable feedback.
>
> We are encouraged that you found our work offers fresh perspectives and will likely stimulate further academic discussion!
>
> Please kindly find the point-to-point response to the questions and suggestions.
>
> ## Major suggestions and questions:
>
> 1. The article is not very clear on how it assesses biological interpretability. a) Is it through the values related to firing rate in the metrics? b) Why would similar firing rates be more biologically similar? c) The methodology for evaluating biological interpretability requires clarification.
> *For example, how model units are aligned or matched with biological neurons? How the neural recordings from LGN were preprocessed?*
> There is no paragraph explaining how the similarity with the LGN regions is calculated and how the neural data is utilized.
>
>     - a) Thank you for pointing this out. In the revised manuscript, we provide a detailed explanation of how the biological interpretability related objectives are computed in Appendix E.1.
>     - b) In this work, we align the neuronal information and connectivity information of the V1 model with biological information. The input stimuli to the V1 model are aligned with the biological visual pathway (i.e., from retina to LGN to V1) by using the LGN model from Billeh et al. (2020) [1] to process the visual input before feeding it into the V1 model [1,2].
>
>         We compare the firing rate value and normalized firing distribution of the V1 model with that of biological V1 neurons to compute the firing rate related objectives (No.5 and No.6 metrics defined in Sec. 2.3). Therefore, similar firing rates indicate that the V1 model is operating under biologically realistic conditions.
>     - c) Please kindly find a comprehensive description of the modelling alignment process and preprocessing steps in "Appendix C".
>
> 1. Many formulas are presented without explaining the meanings of all the symbols. It is recommended to provide a complete explanation.
>
>    - We appreciate the reviewer's observation regarding the clarity of our mathematical notation. In response, we have revised the manuscript to include a summary of important symbols used in our paper in the "Appendix A".
>
> ---
> Reference:
>
> - [1] Billeh, Y. N., Cai, B., Gratiy, S. L., Dai, K., Iyer, R., Gouwens, N. W., ... & Arkhipov, A. (2020). Systematic integration of structural and functional data into multi-scale models of mouse primary visual cortex. Neuron, 106(3), 388-403.
> - [2] Chen, G., Scherr, F., & Maass, W. (2022). A data-based large-scale model for primary visual cortex enables brain-like robust and versatile visual processing. Science advances, 8(44), eabq7592.

---

### Official Review · Reviewer_XrNa · 2025-11-01

**Soundness:** 2
**Presentation:** 3
**Contribution:** 2
**Rating:** 2
**Confidence:** 4

**Summary:**

This paper presents a large-scale framework for evolving heterogeneous synaptic plasticity rules within a model of mouse V1. This work is ambitious, and seeks to unify many observed biological features of synaptic plasticity by using a multi-objective optimization across biological and task metrics and exploring a vast parameter space of rules. The approach is technically impressive and represents a major engineering effort. However, the biological and computational insights drawn from the discovered rules remain limited: the evolved rules are simple, the tasks are minimal, and there is little analysis of why heterogeneity helps.

**Strengths:**

- Ambitious attempt to systematically explore a large rule space under biological constraints.
- Clear and reproducible description of the evolutionary framework.
- A tour de force of synaptic plasticity modelling and related works.

**Weaknesses:**

The scientific hypothesis being tested isn’t clearly stated. Filling in gaps between existing approaches isn’t a hypothesis. I think stemming from this, the paper is somewhat unfocused in its choice of related literature and methods description throughout. For example I think much of section 2 could go in the appendix. As a result, there is no space for details of the V1 model in the main text.

It reads as if many of the modelling choices are included because they exist in the literature and field, rather than being focused behind testing a hypothesis.  For example, it is not clear why the network needs to be so large: if general principles are being discovered, a reduced network would likely be a more tractable and informative place to start than beginning with 17 neuron types. Also, to this point, it is not shown clearly that heterogeneous complexity is even needed. This is especially pertinent given the simplistic task nature, and simplicity of the rank1 and rank2 rules found.

In my opinion the primary weakness that needs to be addressed is the empirical results. These results, as currently presented, do not support the claims in the abstract. The single task (image change detection) is overly simple and does not support the general claim of “high task performance”. Moreover, I am not convinced by the comparison to Adam. From Figure 6, it looks as if there is an initialisation issue with Adam. At the very least there should be a hyperparameter search over Adam’s parameters. It may also be that without some regularisation (e.g. the metrics in 3.3) the network struggles to learn. Also, cutting off training when task accuracy is ~%70 is strange. Neither learning rule appears to have been run to convergence and again, this is important for the claims made.

I would find the claims of superior performance over GD more convincing if the authors were to take an existing SANN task/dataset, for example Heidelberg Digits, and compare their framework to an optimized surrogate gradient benchmark. Ideal would be a small set of tasks that required the network to implement different computational strategies.

Finally, there is minimal mechanistic insight or discussion into how the rules support learning.

**Questions:**

Questions

1. The Adam baseline in Fig. 6 seems poorly tuned and underperforms due to a long period of no learning. Have the authors tested alternative optimiser settings, initialisations, and standard regularisations?
2. Is the heterogenous nature of plasticity rules required? Have the authors tested reduced models with fewer cell types? It also appears like very simple rules are sufficient for the task - is that the correct interpretation?
3. How important is the task choice for the conclusions? Have the authors tested this framework on other tasks?
4. What insight is gained about why certain terms (pre/post, eligibility, reward) occur in the best rules?

Minor questions:
1. What is n in (2)? Also, l_i is a little odd for a trial index?
2. Eq 7, g^k is learned by evolutionary search, correct?
3. I don’t think the implementation of metrics in 3.3 is explained? They might also be suitable as regularisers for GD.
4. Are the authors making an implicit assumption that plasticity rules are learning rules? I think a clear statement or discussion on the relationship between plasticity rules and learning rules could be valuable to the reader (but do not think this is a must). My perspective is that synaptic plasticity rules are often components of a larger learning rule or process. But I’d be interested in their perspective. The authors touch on this in paragraph 393 with the observation that reward free rules succeeded.
5. Dale’s law, maximum & minimum weights, adaptive scaling (appendix B.2) might be valuable aspects that sculpt which plasticity rules are found. It reads as a little odd that these features were chosen to be explicitly dealt with if the whole point is to discover rules that fit biology. Do the authors have a justification?

---

> ### Author Response · Authors · 2025-11-22
> **Response to Reviewer XrNa [1/4]**
>
> Thank you for acknowledging our work in synaptic plasticity modelling and rule discovery.
>
> We also sincerely appreciate your detailed feedback and review!
>
> We hope the following responses will address your concerns.
>
> ## Major concerns and questions:
> 1. Scientific hypothesis being tested isn’t clearly stated.
>
>     We thank the reviewer for this important feedback. We have revised the **Introduction** to clearly articulate the research questions that guide our study:
>     - Question: What are the possible mathematical components and computational principles underlying heterogeneous plasticity rule that can simultaneously satisfy biological constraints and enable functional behaviors in realistic cortical circuits?
>     - Our work takes a `discovery-driven approach` to address this research question. Thus, rather than testing a specific hypothesis, we explore the landscape of possible plasticity rules through multi-objective evolutionary optimization, revealing multiple possible computational strategies that biological systems might employ. We have also added more discussion about the explored plasticity rules with the previous literature in the added **Discussion** section.
>
> 2. Move some description in Sec. 2 to the appendix since no space for details of the V1 model in the main text.
>
>      - We have merged the **Introduction** and **Related Work** sections, creating additional space in the main text accordingly.
>      - As our main contributions are the plasticity rule discovery framework and discovered insights, we allocate the Section 2 to:
>         - Construction of the interpretable search space.
>         - Multi-objective evolutionary optimization methodology.
>         - Evaluation metrics balancing biological plausibility and task performance.
>      - We have provided a concise overview of the visual pathway model (including LGN, LGN-to-V1, and V1 components) in the Sec. 2.2, sufficient for understanding our framework's operation. Comprehensive architectural details, connectivity patterns, and biophysical parameters are documented in Appendix C.
>
>
> 3. Many of the modelling choices are included because they exist in the literature and field, rather than being focused behind testing a hypothesis. a) Why network needs to be so large? A reduced network would likely be a more tractable and informative place to start than beginning with 17 neuron types. b) Why heterogeneous complexity is needed? c) Have the authors tested reduced models with fewer cell types?
>
>     - a) Our goal is to infer what plasticity rule might operate in real biological circuits, rather than achieve high task performance. Starting from a biologically realistic architecture is the core of our methodology. We choose to use a 3000-neuron V1 model because this scale:
>         - maintains sufficient neuronal population size within each of the 17 cell types to exhibit realistic network dynamics and firing rate distributions,
>         - and thus enables statistical comparison with experimental data (e.g., firing rate distributions across cell types).
>
>         Network models with significantly fewer neurons fail to capture the diversity and complexity of interactions.
>     - b) Regarding the reason for bringing heterogeneity into the model:
>       - Follows the established V1 taxonomy with 17 neuron types (see Fig. 1A and 1C in [1]), the V1 model provides the necessary substrate for validating the model's dynamics and properties with the optimization of explored plasticity rules.
>       - This intermediate number of neuron types (17 types) is similar to other research [1-3], which represents a field-recognized granularity balancing biological realism and tractability.
>     - c) We have tested reduced models with 2 types (E/I) and 9 types (layer-wise E/I) while maintaining the same number of neurons. However, we failed to reproduce the diverse dynamics observed in the full model.
>
>         The lack of straightforward mapping between plasticity rules at different granularities is perhaps *unsurprising*, as different cell type granularities can introduce distinct dynamics and interactions influencing overall network function. This finding is also consistent with the "more is different" principle [3].

---

> ### Author Response · Authors · 2025-11-22
> **Response to Reviewer XrNa [2/4]**
>
> 4. a) Is the visual change detection task really difficult and has complexity? b) The single task (image change detection) is overly simple and does not support the general claim of “high task performance”.
>     - a) The visual change detection task isn’t challenging for the network’s capacity, but it is `challenging for the learning rule`, because it requires handling long-range dependencies `without any gradient signal`. Even with the guidance of a gradient signal, BPTT usually fails in capturing long-range dependencies.
>
>         Additionally, the visual change detection task is indeed `challenging for mice`. Experimental studies show that even well-trained mice achieve behavioral performance around ~60% accuracy for grating stimulus [4] and ~73% accuracy for natural image stimulus [5], indicating the task is far from trivial.
>     - b) The two visual change detection tasks have different input distributions: one uses only grating stimuli, while the other uses only natural images.
>
>         The "high task performance" claim refers to the ability of the discovered plasticity rules to enable V1 model to perform the visual change detection task at a level comparable to biological systems (mice), rather than achieving near-perfect accuracy.
>
>         We have clarified this in the revised manuscript.
>
> 5. The comparison to Adam is not convincing enough. a) It seems that the setting of Adam is not well tuned. b) It may also be that without some regularisation (e.g. the metrics in 3.3) the network struggles to learn. c) The authors can test alternative optimiser settings, initialisations, and standard regularisations.
>
>    - We have tuned the hyperparameters for the Adam optimizer indeed, the results reported in the original submission represent the best performance we could achieve after hyperparameter tuning. For a clear comparison, we have added a sensitivity analysis of Adam and SGD hyperparameters (learning rate) in **Appendix** G.
>    - For fair comparison, we have applied the same biological plausibility metrics provided in Section 2.3 as part of the loss during Adam training (except Obj-2, which refers to rule complexity). We have clarified this in the revised Sec. 3.3 and added the training setup details in Appendix G.
>    - Adam sensitivity table: learning rate vs. steps.
>
>     | lr \ Step | 1 | 10 | 100 | 1,000 | 10,000 | 100,000 | 200,000 |
>     |-----------|---------|---------|---------|---------|---------|---------|---------|
>     | 1e-5 | 49.5% | 53.6% | 47.2% | 48.6% | 50.3% | 49.4% | 50.1% |
>     | 1e-4 | 48.5% | 53.5% | 47.2% | 48.6% | 50.3% | 49.8% | 70.2% |
>     | **1e-3** (used) | 52.3% | 53.8% | 47.2% | 48.6% | 50.3% | 90.2% | 98.0% |
>     | 1e-2 | 49.6% | 53.3% | 47.2% | 50.2% | 65.8% | 68.9% | 73.1% |
>    - SGD sensitivity table: learning rate vs. steps.
>
>     | lr \ Step | 1 | 10 | 100 | 1,000 | 10,000 | 100,000 | 200,000 |
>     |-----------|---------|---------|---------|---------|---------|---------|---------|
>     | 1e-5 | 49.5% | 53.6% | 47.2% | 48.6% | 50.3% | 49.4% | 50.1% |
>     | 1e-4 | 48.5% | 53.5% | 47.2% | 48.6% | 50.3% | 54.1% | 55.7% |
>     | 1e-3 | 52.3% | 53.8% | 47.2% | 48.6% | 52.7% | 54.6% | 53.6% |
>     | 1e-2 | 49.6% | 53.3% | 47.2% | 50.8% | 50.5% | 51.6% | 51.4% |
>
> 6. Cutting off training when task accuracy is `~70%` is strange. Neither learning rule appears to have been run to convergence, and again, this is important for the claims made.
>
>    - Our initial intention of cutting off the training at `~70%` was to align with the biological performance of mice (~60% for grating and ~73% for natural images).
>    - We acknowledge that GD can reach higher performance with more training samples in both tasks (See Fig. 13 and **Appendix** G for details).
>
> 7.  a) Suggestion to include more datasets, such as `Heidelberg Digits` to enhance the generality of the findings. b) How important is the task choice for the conclusions? Have the authors tested this framework on other tasks?
>        - a) **1.** Since our framework is constructed based on the V1 model and the input data needs to be passed through the visual pathway model (i.e., LGN + LGN-to-V1 modules), it is non-trivial to apply the framework to other modalities. **2.** Additionally, the multi-objective optimization relies on biological data to constrain the search for plasticity rules. Thus, we cannot directly apply the framework to Heidelberg Digits without corresponding biological data.
>        - b) We acknowledge that the selection of the task setting may lead to different results. We also acknowledge that incorporating additional tasks can strengthen the generality of our conclusions. However, to the best of our knowledge, no other publicly available visual datasets provide comparable neuronal activity data, limiting our ability to test additional tasks at this time.

---

> > ### Author Response · Authors · 2025-11-22
> > **Response to Reviewer XrNa [3/4]**
> >
> > 8.  Add more discussion about mechanistic insights gained from the plasticity rules discovered. What insight is gained about why certain terms (pre/post, eligibility, reward) occur in the best rules?
> >
> >     We have added more mechanistic insights gained from the discovered plasticity rules in the **Discussion** section. In short, we try to cover the following points to provide more insights into the explored rule:
> >
> >     - Functionally equivalent plasticity rules may explain why contradictory mechanisms are observed in experiments.
> >     - Relation between the plasticity rule (both reward-free and reward-required) and memory.
> >     - Relation between the explored plasticity rule and the innate abilities.
> >
> > 9.   It also appears like very simple rules are sufficient for the task - is that the correct interpretation?
> >
> >      - Yes. Please kindly find the detailed explanation in **Discussion** Sec. 4.2 and Sec. 4.3.
> >      - In short, we speculate that this learning efficiency may stem from evolutionary embedding of task-related priors into the plasticity rule structure and parameters (similar to `inductive bias` in AI), since `the task setting is fixed` during the evolution process. Thus, the explored evolved plasticity mechanisms may encode ancestrally relevant computational strategies, enabling rapid adaptation without extensive trial-and-error learning. The evolution process echoes to all species that evolve and interact with the environment in the real world, since many tasks can be treated as `fixed tasks` during the evolution timescale. We call it a `plasticity view` to explain where the `innate ability` comes from.
> >      - Additionally, the simple rules are not isolated. They are working in a cooperative fashion with other priors and constraints provided in the model, such as the Dale's law, weight bounds, adaptive scaling mechanism.
> >
> >
> > ## Minor concerns and questions:
> > 10. What is n in (2)? Also, $l_i$ is a little odd for a trial index?
> >     - In Eq. (2), $N_{win}$ denotes the window size for computing the moving average of recent reward.
> >     - We have provided a notation table in Appendix A to clarify important symbols used in the manuscript. The symbol $l_i$ is not central to the main text; we selected it simply to avoid conflicts with other notation, and it doesn't carry a specific meaning beyond serving as an index.
> >
> > 11. Eq 7, $g^k$ is learned by evolutionary search, correct?
> >     - Yes, $g^k\in \{0,1\}$ is learned by evolutionary search.
> >
> > 12. a) I don’t think the implementation of metrics in 3.3 is explained? b) They might also be suitable as regularisers for GD.
> >
> >     - a) We thank the reviewer for pointing out this omission. We have added a detailed implementation of all 6 metrics in Appendix E.1.
> >
> >         Task performance metrics:
> >         - Obj-1 (Multitask performance): Average accuracy across grating and natural image change detection tasks, computed as the proportion of correct responses (hit during change trials + correct rejection during no-change trials) over 200 test trials per task.
> >
> >         Complexity metric:
> >         - Obj-2 (Rule complexity): Number of active terms, computed as $\sum_{k=1}^{25} g_k$ where $g_k$ are binary selection indicators.
> >
> >         Biological plausibility metrics:
> >         - Obj-3 (Maximum firing rate): Peak instantaneous firing rate across all neurons during task execution, averaged over test trials.
> >         - Obj-4 (Asynchronous firing proportion): Proportion of time steps where simultaneously active neurons exceed 1% of the population, averaged over test trials.
> >         - Obj-5 (Firing rate difference): Mean absolute difference between model firing rates (averaged across neurons and test trials) and experimental mean firing rates from Allen Institute visual behavior datasets [5,6].
> >         - Obj-6 (Firing rate distribution difference): Wasserstein distance between the model's firing rate distribution in test trials and the experimental distribution from [5,6].
> >
> >     - b) They are applied as part of the loss during GD training for fair comparison. See our response to Major Concern 5 for more details.

---

> > > ### Author Response · Authors · 2025-11-22
> > > **Response to Reviewer XrNa [4/4]**
> > >
> > > 13. Are the authors making an implicit assumption that plasticity rules are learning rules? I think a clear statement or discussion on the relationship between plasticity rules and learning rules could be valuable to the reader (but do not think this is a must). My perspective is that synaptic plasticity rules are often components of a larger learning rule or process. But I’d be interested in their perspective. The authors touch on this in paragraph 393 with the observation that reward-free rules succeeded.
> > >
> > >     We thank the reviewer for this insightful question about the relationship between plasticity rules and the learning process!
> > >     - The success of `reward-free rules` was unexpected and revealing. We deduce that these rules succeed not because they independently "discover" the task solution from scratch, but because evolution has pre-configured them with appropriate inductive bias provided by the environment (i.e., fixed task setting in this work).
> > >     - While we initially treated the explored rules as a complete learning mechanism, our experimental results suggest they may function as components of a broader learning system rather than operating in isolation.
> > >     - Please kindly find the detailed possible explanation in the **Discussion** Sec. 4.3.
> > >
> > >
> > > 14. Dale’s law, maximum & minimum weights, adaptive scaling might be valuable aspects that sculpt which plasticity rules are found. It reads as a little odd that these features were chosen to be explicitly dealt with if the whole point is to discover rules that fit biology. Do the authors have a justification?
> > >
> > >     - We appreciate the reviewer highlighting the importance of these biological constraints in shaping discovered plasticity rules.
> > >
> > >     - We take these constraints as inductive biases that guide the evolutionary search towards biologically plausible solutions. From biological and modeling perspectives, these constraints serve the following roles:
> > >       - Dale's law reflects a fundamental biological constraint that each neuron releases either excitatory or inhibitory neurotransmitters, but not both. Enforcing Dale's law in our model ensures that the discovered plasticity rules operate within biologically realistic synaptic architectures, enhancing the plausibility of our findings.
> > >       - Maximum and minimum weight constraints prevent synaptic weights from reaching unphysiological extremes, which could lead to unrealistic network dynamics. We adopted these bounds in our work to help maintain stable activity patterns and ensure that the plasticity rules evolve within a feasible parameter space.
> > >       - Adaptive scaling mechanisms help regulate synaptic strength, preventing runaway excitation or inhibition. This homeostatic regulation is critical for maintaining functional balance in neural circuits, and incorporating it into our model allows the discovered plasticity rules to reflect this essential biological process.
> > >
> > >     - From an engineering perspective, consider that learnable constraints of the above aspects can lead to the following issues:
> > >
> > >       - If not explicitly enforced the above constraints, in the initial step of evolutionary search, synaptic weights usually frequently violate this principle, leading to a population-level failure to converge towards biologically plausible solutions.
> > >
> > >       - These rules are non-linear since we need to do the clipping operation on the synaptic weights. Therefore, this will significantly expand the search space, making it challenging for the evolutionary algorithm to efficiently explore and identify effective plasticity rules.
> > >
> > > ---
> > > Reference:
> > >
> > > [1] Billeh, Y. N., Cai, B., Gratiy, S. L., Dai, K., Iyer, R., Gouwens, N. W., ... & Arkhipov, A. (2020). Systematic integration of structural and functional data into multi-scale models of mouse primary visual cortex. Neuron, 106(3), 388-403.
> > >
> > > [2] Campagnola, L., Seeman, S. C., Chartrand, T., Kim, L., Hoggarth, A., Gamlin, C., ... & Jarsky, T. (2022). Local connectivity and synaptic dynamics in mouse and human neocortex. Science, 375(6585), eabj5861.
> > >
> > > [3] Anderson, P. W. (1972). More is different: broken symmetry and the nature of the hierarchical structure of science. Science, 177(4047), 393-396.
> > >
> > > [4] Glickfeld, L. L., Histed, M. H., & Maunsell, J. H. (2013). Mouse primary visual cortex is used to detect both orientation and contrast changes. Journal of Neuroscience, 33(50), 19416-19422.
> > >
> > > [5] Garrett, M., Manavi, S., Roll, K., Ollerenshaw, D. R., Groblewski, P. A., Ponvert, N. D., ... & Olsen, S. R. (2020). Experience shapes activity dynamics and stimulus coding of VIP inhibitory cells. elife, 9, e50340.
> > >
> > > [6] Siegle, J. H., Jia, X., Durand, S., Gale, S., Bennett, C., Graddis, N., ... & Koch, C. (2021). Survey of spiking in the mouse visual system reveals functional hierarchy. Nature, 592(7852), 86-92.

---

> > ### Comment · Reviewer_XrNa · 2025-11-26
> >
> > I appreciate the authors’ detailed clarifications and will read and respond to the full revisions in further detail shortly. Before that, and in the interest of time, I want to highlight two points that I believe remain unresolved and may still be addressable.
> >
> > Regarding comment 5.
> > The added Adam sensitivity analysis explores only the learning rate. However, for Adam and SGD, the momentum parameter (SGD) and $\beta_1$ / $\beta_2$ parameters (Adam) often strongly determine stability and sample efficiency (as well as initialisation details). Restricting the search to learning rate alone is, in my view, insufficient to establish a fair comparison. I encourage the authors to widen the hyperparameter sweep (at least for adam), as the present results do not yet convincingly demonstrate that gradient-based methods have been given an appropriately tuned baseline.
> >
> > Regarding comment 6.
> > I appreciate the authors’ revised emphasis on behavioural outcome. For clarity, could the authors report what performance the best-performing discovered rule achieves when trained with as many samples as Adam requires to reach 90% / 98% accuracy? From Fig. 13 it appears that, assuming no change of learning regime, the best rule might match or exceed Adam’s performance by ~$10^5$ updates. As it stands, the discovered rule seems to be stopped early, prior to convergence, making the comparison difficult to interpret and build on in future work.

---

> > > ### Author Response · Authors · 2025-12-01
> > > **Response to further concerns by Reviewer XrNa**
> > >
> > > 1. Requirement to tune 2 momentum $\beta_1$ and $\beta_2$ hyperparameters for Adam on V1 model.
> > >
> > >     We sincerely appreciate the feedback and highlighting the importance of the gradient-based optimizer configuration. We understand the concern that momentum parameters in Adam/SGD can influence stability and sample efficiency.
> > >
> > >     We have conducted additional experiments to assess the impact of tuning $\beta_1$ and $\beta_2$ hyperparameters on the performance of Adam optimizers. Specifically, we performed a grid search over a range of $\beta_1$ (0, 0.5, 0.999) and $\beta_2$ (0.99, 0.999) for Adam. Please kindly find the results in the revised `Fig. S15b`.
> > >
> > >     The results indicate that while tuning these momentum hyperparameters can lead to some improvements in convergence speed, the overall data efficiency gap between Adam and the discovered plasticity rules remains substantial. The best-performing Adam configuration still requires `orders of magnitude more samples` to reach behaviorally comparable performance levels, as shown in `Fig. 6b` and `Fig. S13b`.
> > >
> > >     We also want to reclarify several points regarding the gradient-based optimizers comparison experiments:
> > >
> > >     1. Our intention with the Adam optimizer is NOT to claim algorithmic superiority over state-of-the-art gradient optimizers, but to mechanistically illustrate the `origin of biological learning efficiency`. The comparison highlights a 'synaptic plasticity view' of `innate abilities`. Unlike GD, which learns from scratch, evolution encodes ancestral strategies into plasticity rules. This suggests innate capabilities arise not just from hardwired circuitry, but from pre-configured plasticity mechanisms that render specific behaviors accessible with minimal experience, bypassing the extensive trial-and-error usually required.
> > >     2. In line with our intention (previous point), our network initialization is based on biological measurements, rather than tuned for GD, unlike typical ANN or SNN initializations that are explicitly designed to suit gradient optimizers. While this biologically based initialization may be suboptimal for GD, tuning it for GD would contradict the purpose of our comparison, so we intentionally keep it as is.
> > >     3. While tuning momentum parameters in a more fine-grained way may improve convergence speed by a constant factor, it is theoretically implausible that such tuning would bridge a 3-to-4 orders of magnitude gap in data efficiency offered by the discovered plasticity rules, as shown in `Fig. 6b` and `Fig. S13b`. [1] clarifies that the momentum parameters of $\beta_1$ and $\beta_2$ in Adam only deteriorate the value of the convergence rate bound (i.e., changes the constant term), but do not change the core convergence order. Please kindly find a detailed proof and discussion in `Sec. 4.2`, `Theorem 3` and `Theorem 4` in [1]. Similar arguments also hold for SGD [2].
> > >
> > > 2. Report what performance the best-performing discovered rule achieves when trained with as many samples as Adam requires to reach 90%.
> > >
> > >     - We appreciate the suggestion to evaluate the best-performing discovered rule with an extended number of training samples. We have conducted additional experiments where we apply the best-performing discovered plasticity rule for `1,000`, `10,000`, `100,000` and `200,000` samples, matching the sample size required by Adam to reach over 90% accuracy. Please kindly find the results in the revised `Fig. S15`.
> > >
> > >     - The results demonstrate that the overall best-performing discovered plasticity rule maintains behaviorally similar performance (`~70%`) even with extended training samples. Specifically, the performance remains stable and does not degrade over time, indicating that the discovered plasticity rule is capable of sustaining effective adaptation without collapsing. This further underscores the efficiency and robustness of the discovered plasticity mechanisms in facilitating learning with minimal experience.
> > >
> > >
> > > ---
> > > **Reference**
> > > - [1] Défossez, A., Bottou, L., Bach, F., & Usunier, N. (2022). A Simple Convergence Proof of Adam and Adagrad. Transactions on Machine Learning Research.
> > > - [2] Nemirovski, A., Juditsky, A., Lan, G., & Shapiro, A. (2009). Robust stochastic approximation approach to stochastic programming. SIAM Journal on optimization, 19(4), 1574-1609.

---

### Official Review · Reviewer_N15c · 2025-11-01

**Soundness:** 3
**Presentation:** 3
**Contribution:** 3
**Rating:** 6
**Confidence:** 4

**Summary:**

This paper proposes to find candidate plasticity rules that could be implemented in the brain using numerical optimisation (meta-learning, “learning the learning rule”). This is an important problem, as direct recordings of weights in vivo during learning are currently impossible, thus computational methods are the only option to integrate all the indirect evidence at our disposal (neural activity during learning, general circuit knowledge, task performance…).

The authors consider a biologically detailed model of mouse V1 cortex inspired by previous work with 289 synapse types, each evolving with their own, tunable plasticity rule. Plasticity rules are parameterised as Taylor expansions up to the third order on the presynaptic activity, postsynaptic activity and reward. To navigate this high-dimensional search space (~2.6k parameters in total), they use a multi-objective evolutionary strategy, refined from previous work that maintains and evolves a population of 4000 rules.

They apply this method to a working memory task (image change detection), for which there exists mouse recordings during learning. They select rules based on several objective: task performance, a bias towards simpler rules (fewer non-zero terms), matching of the in silicon and in vivo neural activity, and general biological plausibility (e.g. physiological weight values, individual firing rates). Each candidate plasticity rule (i.e. 289 individual rules) is evaluated by exposing the network to a 100 trials with plasticity on, while performance is evaluated on test trials with plasticity turned off. This means that task needs to be performed with a fixed connectivity learned by plasticity during training, and not via online weight changes during individual trials.

They find rules that solve the task with comparable performance to the animals. They show that the rules are scale-invariant via simulations in smaller and bigger networks of similar architecture. Their set of 70 best-performing rules display similar fit to data yet meaningfully different expressions (degeneracy). Notably, non-reward modulated rules, and simple, pre-only rules can solve the task, showing that the task at hand can be solved in an unsupervised way.  Finally, they show that their best rules learn the task faster than an implementation of gradient based optimization (Adam).

**Strengths:**

- The method presented has many strong points compared to existing work: (i) flexible, high dimensional search space with co-active rules across many synapse types, (ii) large scale, biologically detailed network model which enables (iii) direct comparison with experimental data (iv) multi-objective optimisation which allows to select meaningful candidate rules (v) some level of a population view: recovering several candidate rules, reporting that many different rules are equally consistent with data. Though there exists work with the ingredients mentioned above, this is, to my knowledge, the first study to put all of these together. The only work that comes close, [2], is 1/ very recent and 2/ trades off the flexibility of the search space for a more exhaustive survey of the space using numerical inference instead of local optimisation, and thus provides a complementary instead of redundant perspective. Thus this work is novel and interesting as it adresses many limitations of existing work, and worthy of being shared with the community.

- This work is a technical tour de force, the sheer number of simulations of large-scale, biologically detailed models with so many co-active rules simulated in parallel on GPUs using modern ML libraries lays the groundwork for the field to embrace such large scale simulations more easily in the future. Navigating and optimising such a landscape is notoriously hard, with most previous work effectively failing to do so. The method presented here appears to work at a scale large enough for direct experimental comparison. Moreover, the biological model, search space and optimisation procedure are very well explained. This is remarkable, given the complexity of the method with nested loops of learning systems and evaluations. Overall, I see the methodological and technical side of this work as exceptionally strong, and the evidence that this method could be useful for neuroscience is convincing (provided a few clarifications and additional plots, see question 1).

**Weaknesses:**

However, despite the exciting aspects above (for which the authors should be commended), I found the results drawn from the method currently lackluster and underdeveloped, especially given how promising their setup is.

1/ The paper lacks a discussion and limitations section, and at present is convincing only as a method paper, not as much as a work that attempts to make actual predictions on which plasticity rules are in the brain. I could still recommend this paper for acceptance for the methods alone, provided some clarifications and some claims to be adjusted accordingly. A much more exciting route would be to dig deeper on the current results and extract some new neuroscientific insights (more on this, with hopefully actionable items in question 3).

2/ Position wrt prior work: though the section comparing this work to prior work is thorough, I think it needs more precision and nuance. Right now, many different papers are bunched together with general statements on limitations for which the reader needs to investigate applies to which reference mention all potential weakness. e.g. for the paragraph line 115: consider a small search space (….), or simple tasks (…). I also think 2 important papers are missing: [1] which is also evolutionary search in an effectively infinite dimension search space, though it is applied to ~single neuron spiking models and simple tasks, without direct comparison to experimental data. [2] which considers 4 co-active rules and a smaller search space, but also in large scale spiking models, with direct comparison to data and a multi-objective selection of rules. Importantly, this paper also finds degeneracy of rules, which is interesting to relate somewhere in discussion to the current paper’s claims. To summarise: 1/ the precise method used in the paper is novel 2/ there exists work that has each of the advantages ascribed by the method, but not all at once. 3/ the combination of the advantages allows for unique capabilities of the method presented (e.g. visualize the trade-offs between different objectives).

3/ Overclaims and paper’s lack of a main message.
- Line 53 (and similarly Line 58): “systematically”: I don’t think that the authors can claim that at present they are systematically exploring the 2.6K parameter space with an evolutionary strategy, almost by design given how big this space is. It’s both a strength (large, very flexible space) and a weakness (hard to survey systematically) of the paper. Despite this limitation that should be acknowledged somewhere, I think it’s very nice that this paper still makes a point about the observed degeneracy. I am pointing this out mainly in contrast to the SBI line of work [2] which infers posteriors over rules to be able to more systematically explore the search space and degeneracy. In their case, the search space has to be much smaller dimensional.
- Can you really call this a “multitask setting”, when it’s 2 tasks, both image change detection? Why not simply image detection task with either natural images or gratings. Currently this makes the rules inferred look like they solve a wider range of tasks than they probably do, and makes especially the simple rules found seem suspicious.
- The comparison with gradient descent. Line 24: “Experimental results demonstrate that the discovered plasticity rules can attain high task performance with orders of magnitude fewer training samples than the gradient-based rule”. I think this statement is vastly overinflated, and I’m not sure how useful this point is, given how underdeveloped the analysis of the results is otherwise. Is the claim destined to alert machine learning practicioners that this paper proposes rules more efficient better than gradient descent? Then many additional results are required to show the results generalises across different tasks and architectures. The gradient descent claim can be rephrased into a more fair sentence acknowledging this is probably heavily dependent (overfit?) on the task and network model considered here (and would require more results on the GD hyperparameter search). I’m not sure of the value of adding the gradient descent debate into this paper, and would rather see the space used to analyse and reverse engineer more thoroughly the rules found to make predictions for neuroscience (see questions). Overall, I think this GD point risks being a distractor on what your framework offers, and I would keep this paper “AI4neuro”, focused at understanding the rules in the brain, and not “Neuro4AI”, aiming to develop novel learning rules for ML inspired by neuro, at least in this submission.
- The final section of the paper brushes too fast over the interpretation of the plasticity rules and implications for neuroscience (see questions).

**Questions:**

(1) Fig 4, I would add a panel about the optimisation itself, not only after convergence (for instance the performance of the best rule at each step). I’m especially curious wrt how no plasticity performs on the task, as well as randomly drawn rules from the search space (step 0 of the optimisation). Right now, this is crucial evidence missing to show that your method efficiently navigates the space of rules. It would be a red flag that most rules in the search space are good at this task.

(2) How reproducible is the optimization run shown in the study? I understand that within one run, you keep a population of rules (4000), but this is very small wrt to the dimensionality of the whole search space, so I would expect that you are “missing” many good candidate rules. For instance, starting from a different initialisation, does a run converge to a completely different population of rules? Or do you see regularities within the inferred rules? Apologies if I missed this experiment. And I understand this is a lot of compute to ask for. But at least this should be mentioned in limitations or in the discussion section. This method is not really fit to properly explore degeneracy (given the dimensionality of the space, probably no method can, but the claims in the paper (line 283) imply the opposite).

(3) I would appreciate more work on analysing the solutions obtained by the optimisation. How come simple rules (pre-only or reward-free rules) are able to solve the task by themselves? How are the networks able to keep a memory of the previous stimulus? Since weights are frozen during testing, it has to be in the activity (or is there short term plasticity in the model?)? Can you decode the identity of previous stimulus from the activity? How does this change across time? Can you compare how a static network (without plasticity) behaves in this task, and what changes do a successful plasticity rule brings, especially the simplest ones you found in the optimization? If the evaluation trials are indeed a good metric of generalization performance, this would mean there is a connection between extremely simple, unsupervised rules and generic working memory abilities, which is exciting, and to my knowledge, novel. Can you analyse and compare the solutions that use/don’t use reward, and make predictions on how we could distinguish these two in future experiments. But as it stands, your results instead question the task choice, if reward is not necessary to learn. My worry is that the advantage of using such a complex search space is lost if simple, reward-free rules can solve the task. At least this warrants proper reverse engineering efforts to understand what these rules do to the network and how they solve the task. Nice insights for neuro could fall from this.

Smaller points:

- Why shut off plasticity during test trials? Presumably the candidate rules are restricted to be slow, or at least comparable to the brain. If you leave plasticity enabled during testing, do you learn radically different plasticity rules? Can the task be fairly recasted as novelty detection? In that case, it doesn’t have to be working memory and it would be a nice link between the working memory literature and the novelty detection literature.

- Line 426: distributions of firing rates on the violin plots do look quite different to data still (mainly, a larger variance in exp data). Normally that wouldn’t be a concern for a network of identical neurons like are standard in the field, but given your much more refined model with many neuron types, how do you interpret the inability to better fit the distribution of firing rate (which is objective 6, so part of the fitting procedure if I understand correctly).

- Define heterogenous: when reading initially I understood heterogenous as one rule per synapse, not one per synapse-type. Maybe co-active is better suited?

- Fig 2: I would say from where the data was taken in the caption, not only in the main text.

- I’m confused by the 2 timescales for reward: one is the window over which past rewards are averaged (20, not learned), and the other is the time constant in eq 4. Why not have a single timescale? And if 2, why not learn both?

- Limitation section: besides the points raised above, I would acknowledge there that in experiments, people are moving away from pairwise rules with for instance btsp [4], which are not taken into account by your model.

- Introduction has a whole paragraph on related work, followed by a section entirely dedicated to related work. Maybe merge the two for space, now it functions a bit like a repeat.

- Line 55: “broad but interpretable candidate space of heterogeneous plasticity rules” Are Taylor expansions really a guarantee of interpretability? Especially when there are so many terms and so many co-active rules? By interpretable I mean that can give rise to theoretical insights or experimental predictions. This is a more general point, so does not affect my assessment of the paper.

- Can you explain the choice for a single g for all synapses (which I understood to be tunable)? Wouldn’t the simplicity criterion introduced in the optimization already remove unnecessary terms, and the g could only be there to remove the non-sensical terms in the Taylor expansion once and for all (and potentially removed from the main text for clarity)?

- Enforcing Dale’s law and maximum values: how often do your weights attempt to cross 0? I.e. how often is this condition not met by the rule itself but instead by your added bound. Same question for the upper bound. Ideally these bounds are not reached often.

-  Given that one of the key conclusion is about degeneracy, a discussion of other work studying degeneracy should be added, for instance [2].

- Please reformat the citations so that they are either in parenthesis or numbered (if ICLR allows this).

- This is not really a question, but I found the scale-free result very nice and reassuring on the solutions that you find.

[1] Jordan et al, Evolving interpretable plasticity for spiking networks, eLife, 2021.
[2] Confavreux et al, Memory by a thousand rules: Automated discovery of multi-type plasticity rules reveals variety & degeneracy at the heart of learning, bioRxiv, 2025.
[3] Bittner et al, Behavioral time scale synaptic plasticity underlies ca1 place fields, Science 2017

---

> ### Author Response · Authors · 2025-11-22
> **Response to Reviewer N15c [1/4]**
>
> Thanks for your careful review and comments.
>
> Glad to see that you highly recognize the methodological contributions of our work and deem our work worthy of being shared with the community!
>
> We have strengthened the result and discussion section according to your comments. Please kindly find our responses below and the revised manuscript.
>
> # Major suggestions and questions:
> 1. The paper needs a discussion and limitations section.
>
>    We have added a **Discussion** section and moved the relevant content from the **Experiments and Results** section to the **Discussion** section. We have also included a **Limitations and Future Work** subsection in the **Discussion** section to further outline its boundaries.
>
>    - Regarding **Discussion** section, we try to cover the following points to provide more insights:
>      - Functionally equivalent plasticity rules may explain why contradictory mechanisms are observed in experiments.
>      - Relation between the plasticity rule (both reward-free and reward-required) and working memory.
>      - Relation between the explored plasticity rule and the innate abilities.
>      - Limitations and future directions.
>
>
> 2. The paper needs to position itself with respect to the following prior work [1,2] clearly.
>
>    Thank you for sharing these two works as well as the discussion logics with us. We have revised the manuscript accordingly. A brief comparison with these two works is provided in the **Discussion** Sec. 4.1 and Appendix B2.
>
> 3. Words overclaimed: "systematically" and "multitask setting"
>    - Remove `systematically`: Your understanding is correct. Systematically exploring such a high-dimensional space is computationally intractable. We have removed this word in the revised version.
>    - Revise `multitask setting`: We have revised the term 'multitask setting' to `cross-domain task setting` throughout the manuscript to avoid potential misunderstanding. Please refer to the Sec. 3.1 in the revised manuscript for more details.
>
> 4. The comparison with gradient descent (GD) is not very convincing. Doubt about the value of adding the GD debate, as it risks being a distraction from what the framework offers.
>
>     We have provided a detailed clarification of the *motivation* for including the `GD` comparison in the **Experiments and Results** section and **Discussion** section to avoid potential misunderstanding. Here is a summary:
>
>    - The GD comparison is NOT intended to show `general superiority` of explored plasticity rules over BP for algorithmic applications. Rather, we try to show the neuroscientific value of our framework via answering a scientific question: *Why do biological systems (e.g., humans) equipped with synaptic plasticity rules exhibit few-shot learning capabilities that differ qualitatively from artificial networks trained via GD?*
>    - Our results reveal an asymmetry: Evolved plasticity rules achieve task performance comparable to real biological systems (~60% for grating [4] and ~73% for natural images [5]) using only hundreds of training samples, while GD-trained V1 model require *orders of magnitude more training samples* to reach similar performance levels under the same optimization constraints (except Obj-2 which is the rule simplicity objective).
>    - We speculate that this learning efficiency may stem from evolutionary embedding of task-related priors into the plasticity rule structure and parameters (i.e., `inductive bias` in AI), since `the task setting is fixed` during the evolution process. Thus, the explored evolved plasticity mechanisms may encode ancestrally relevant computational strategies, enabling rapid adaptation without extensive trial-and-error learning. We have added detailed discussion in the revised **Discussion** Sec. 4.3.
>
> 5. Has the Adam optimizer been tuned? Have we tried other optimizers?
>    - Yes, we have tuned the Adam optimizer. We have also tried the `SGD` optimizer.
>    - We have added two hyperparameter analysis in Appendix G.
>    - Adam sensitivity table: learning rate vs. steps.
>
>     | lr \ Step | 1 | 10 | 100 | 1,000 | 10,000 | 100,000 | 200,000 |
>     |-----------|---------|---------|---------|---------|---------|---------|---------|
>     | 1e-5 | 49.5% | 53.6% | 47.2% | 48.6% | 50.3% | 49.4% | 50.1% |
>     | 1e-4 | 48.5% | 53.5% | 47.2% | 48.6% | 50.3% | 49.8% | 70.2% |
>     | **1e-3** (used) | 52.3% | 53.8% | 47.2% | 48.6% | 50.3% | 90.2% | 98.0% |
>     | 1e-2 | 49.6% | 53.3% | 47.2% | 50.2% | 65.8% | 68.9% | 73.1% |
>    - SGD sensitivity table: learning rate vs. steps.
>
>     | lr \ Step | 1 | 10 | 100 | 1,000 | 10,000 | 100,000 | 200,000 |
>     |-----------|---------|---------|---------|---------|---------|---------|---------|
>     | 1e-5 | 49.5% | 53.6% | 47.2% | 48.6% | 50.3% | 49.4% | 50.1% |
>     | 1e-4 | 48.5% | 53.5% | 47.2% | 48.6% | 50.3% | 54.1% | 55.7% |
>     | 1e-3 | 52.3% | 53.8% | 47.2% | 48.6% | 52.7% | 54.6% | 53.6% |
>     | 1e-2 | 49.6% | 53.3% | 47.2% | 50.8% | 50.5% | 51.6% | 51.4% |

---

> > ### Author Response · Authors · 2025-11-22
> > **Response to Reviewer N15c [2/4]**
> >
> > 6. Add a panel about the optimization (for example, the performance of the best rule at each step).
> >    - We have added a detailed analysis of the search space in **Appendix** E.3 (including Table 3). The search space is highly complex, and random selection most likely yields non-functional rules.
> >    - Regarding the optimization trajectory, since we simultaneously optimize six objectives, a single "best" metric per step is ill-defined. We believe the comparison between the baseline (**Appendix** E.3) and the final results demonstrates the method's efficiency in navigating this space.
> >
> > 7. How reproducible is the optimization run shown in the study?
> >    - We have run several independent optimization runs with different random seeds. The results are almost consistent across different runs. We have added a note in the **Experiments and Results** section to clarify this point.
> >    - However, we need to point out that the selection of the evolutionary algorithm for optimization and model setting should lead to different exploration results.
> >
> > 8. How come simple rules (reward-free rules) are able to solve the task by themselves?
> >    - We speculate that this may stem from the evolutionary process. Since the `task setting is fixed`, the explored evolved plasticity mechanisms may encode the task information within the rule structure and parameters during the evolution process, enabling rapid adaptation without extensive trial-and-error learning. We have added this discussion to the **Discussion** Sec. 4.3.
> >
> > 9. **[Working memory questions]** a) How are the networks able to keep a memory of the previous stimulus? b) Can you decode the identity of the previous stimulus from the activity? How does this change across time?
> > c) Can you compare how a static network (without plasticity) behaves in this task? d) What changes does a successful plasticity rule bring, especially the simplest ones you found in the optimization? Can you analyse and compare the solutions that use/don’t use reward? e) Can you make predictions on how we could distinguish these two in future experiments?
> >
> >      Thanks for raising these interesting questions! We have revised the **Results** and **Discussion** sections accordingly. Please find our responses below:
> >
> >      - a) From Fig.5 in the manuscript, we can see that the different types of neurons exhibit different firing patterns during the delay period between the current trial's response window and the next input. This is a direct sign that the network is able to keep a memory through this spike train [6]. We have provided layer-wise dynamics in Fig. 10 for better visulization.
> >      - b) Yes. Since our task is a visual change detection task, the network needs to compare the current stimulus with the previous one sequentially. Thus, the identity of the previous stimulus should be able to be decoded from the activity of the network during the response period.
> >      - c) We have provided this analysis in Fig. 6b and Fig. 13b. When training sample $N=0$, the decoding accuracy of the static V1 network is at a 50/50 level. As the training sample increases, the decoding accuracy also increases, indicating that the network can keep a memory of the latest two stimuli and ignore the older ones.
> >      - d) We have provided the dynamics of performance gains brought by 3 different plasticity rules across different numbers of training samples in Fig. 6b and Fig. 13b. The yellow line ($\Delta w = S_{pre}$) and green line ($\Delta w = S_{pre}\cdot X_{post}$) indicate the top two prevalent rules in the filtered rule population, respectively. Note that both rules are `reward-free` rules. The red line indicates the best-performing rule, which is a `reward-required` rule ($\Delta w = X_{post} + S_{pre} \cdot X_{pre} + S_{post} \cdot X_{pre} + X_{post}^2 + X_{post} \cdot R$).
> >      - e) Please kindly find our detailed discussion in the revised Sec. 4.2 and Sec. 4.3.
> >      Based on our analysis, we can make the following predictions for future experiments:
> >         1. If the task is fixed (i.e., environment remains stable) during the evolution process, then reward-free plasticity rules and reward-required plasticity rules can both lead to the emergence of working memory capabilities for the network and then solve the task. This hypothesis has been partially verified in our current experiments to some extent.
> >         2. However, if the task is changing (i.e., the environment evolves) during the evolution process, we deduce that plasticity rules with reward will dominate. This is because the reward signal can provide information about the evolving task structure and guide the learning process, whereas simple reward-free rules cannot. This hypothesis can be tested in future experiments.

---

> > > ### Author Response · Authors · 2025-11-22
> > > **Response to Reviewer N15c [3/4]**
> > >
> > > # Minor suggestions and questions:
> > > 10.  Why shut off plasticity during test trials?
> > >
> > >      - We shut off the plasticity during test trials due to the following two reasons:
> > >        1. [Evaluation friendly]: We want to evaluate whether the optimized V1 model can perform well after a specific number of training trials with the explored plasticity rule. Turning off plasticity during test trials can prevent further adaptation of the network and thus provide stable evaluation results.
> > >        2. [Computation friendly]: Turning off plasticity during the test phase can save almost half of the computational power. The V1 model adopted in this work has 3,000 neurons, which results in more than 400,000 synapses to be updated at each time step. Thus, turning off plasticity during the test phase in our setting can significantly reduce the computational cost of the evolutionary optimization.
> > >      - One can do *test time training* as well. But we think this will not change the main results of this work.
> > >
> > > 11.  The distributions of firing rates (Fig.5) on the violin plots do look quite different to the data still. In a homogeneous neuronal model, the firing rate distribution should be easy to align well.
> > >
> > >         We thank the reviewer for this important observation. We acknowledge that discrepancies exist between the firing rate distributions in our model and experimental data, particularly in variance. This difference stems from several factors:
> > >      - Model complexity: As the reviewer notes, our model incorporates 17 distinct neuron types. This brings more biological details than homogeneous models, while it also increases optimization difficulty in firing rate dynamics matching.
> > >      - Network scale: The sampled V1 model contains only 3,000 neurons, which is orders of magnitude smaller than the actual V1 area in mice (millions of neurons). This scale difference can impact firing rate statistics.
> > >      - Multi-objective trade-offs: Our framework balances 6 competing objectives simultaneously (see Sec. 2.2 and 2.3). Firing rate matching in terms of distribution (Obj-5) and scale (Obj-6) represents two objectives that must be traded off against task performance and other biological constraints.
> > >      - Temporal compression: For computational tractability during our large-scale evolutionary search, we compressed the simulation timeline to ~150 ms per trial, which is shorter than experimental protocol duration (from ~500ms to ~4s for grating change detection [4] and 750 ms for natural image change detection [5]). This can be an important factor affecting firing rate statistics.
> > >
> > >         We have provided a brief discussion in the revised **Discussion** section. Despite these statistical misalignments, the optimized V1 model is still able to capture key qualitative features: the heavy-tailed distribution and low mean firing rates, which make our results still compelling.
> > >
> > > 12.  Define `heterogeneous` more clearly. The reader may misunderstand it as one rule per synapse, rather than one rule per synapse type. `Co-active` is better suited?
> > >
> > >      - We have revised the definition of "heterogeneous" in the **Introduction** to explicitly state that it refers to synapse-type-specific plasticity rules.
> > >      - Regarding terminology choice: We retain `heterogeneous` rather than `co-active` to emphasize a key distinction from prior work. The similar term `co-dependent` has been established in [7], making "heterogeneous" a clearer choice to avoid potential confusion.
> > >
> > > 13.  Add the information about where the data was taken in the caption of Fig.2.
> > >
> > >      - We have revised the caption of Fig.2 accordingly.
> > >
> > > 14. Confusion of the 2 timescales for reward: one is the window over which past rewards are averaged (20, not learned), and the other is the time constant in Eq. 4. Why not have a single timescale? And if 2, why not learn both?
> > >
> > >     - The two timescales serve distinct roles: the fixed window, $N_{win}$, defines the temporal horizon for computing reward prediction errors (Eq. 2-3), while the learned decay constant $τ_R^m$ (Eq. 4) captures neuron-type-specific neuromodulatory dynamics observed experimentally [8].
> > >     - We chose to learn only $τ_R^m$ since the fixed window provides a stable reference frame while learned decay constants capture heterogeneous neuromodulation across cell types.
> > >     - We acknowledge that jointly optimizing both timescales could yield improvements. But discussing the *dynamics of temporal credit assignment* is beyond the scope of this work.
> > >
> > > 15. Limitation section: besides the points raised above, I would acknowledge that in experiments, people are moving away from pairwise rules with for instance, BTSP [3], which are not taken into account by your model.
> > >     - We have discussed this point in the **Discussion** Sec. 4.4.

---

> > > > ### Author Response · Authors · 2025-11-22
> > > > **Response to Reviewer N15c [4/4]**
> > > >
> > > > 16. The introduction has a whole paragraph on related work, followed by a section entirely dedicated to related work. Maybe merge the two for space; now it functions a bit like a repeat.
> > > >     - We have merged them accordingly in the revised version.
> > > >     - The dedicated related work is moved to Appendix B.
> > > >
> > > > 17. Does the Taylor expansion guarantee `interpretability`? Concerns about whether it can give rise to theoretical insights or experimental predictions.
> > > >     - Regarding "interpretable", we define it in this paper as each plasticity rule takes an `explicit mathematical form` where individual terms correspond to specific neural mechanisms (e.g., pre-synaptic activity, eligibility traces, reward signals) with learnable coefficients.
> > > >     - This contrasts sharply with black-box approaches (e.g., neural networks parameterizing plasticity rules), where the mapping from neural signals to weight changes remains opaque. Our Taylor expansion framework ensures that discovered rules can be directly `inspected`, `analyzed`, and `compared` to experimental observations.
> > > >
> > > >
> > > > 18. a) Can you explain the choice for a single $g$ for all synapses (which I understood to be tunable)? b) Wouldn’t the simplicity criterion introduced in the optimization already remove unnecessary terms, and the $g$ could only be there to remove the non-sensical terms in the Taylor expansion once and for all (and potentially removed from the main text for clarity)?
> > > >
> > > >     - a) The binary indicator $g_k$ is shared across all synapse types for each candidate term $P_k$ to enforce structural consistency in plasticity rules. This shared architecture enables our framework to discover families of plasticity rules with common computational motifs.
> > > >     - b) While the simplicity objective (Obj-2) penalizes complexity, the binary $g$ serves a complementary but distinct role from the engineering perspective:
> > > >         - Discrete vs. continuous optimization: The simplicity criterion operates on the number of active terms, but coefficient optimization alone cannot definitively determine term inclusion/exclusion. The binary $g$ provides explicit discrete structure selection.
> > > >         - Computational efficiency: Pre-filtering terms via $g$ reduces the active search space during evolution, accelerating convergence compared to relying solely on coefficient-based regularization.
> > > >
> > > > 19. Enforcing Dale’s law and maximum values: how often do your weights attempt to cross 0 and the upper bound? Ideally, these bounds are not reached often.
> > > >
> > > >     - We have added a discussion about the weight update constraints in **Appendix** D.2.
> > > >
> > > > 20. Add a discussion of other work studying `degeneracy` of plasticity rules, for instance [2].
> > > >     - We have added this work in the **Discussion** Sec. 4.1 accordingly.
> > > >
> > > > 21. Reformat the citations in parentheses or numbered style.
> > > >     - We have revised and reformatted the citations according to the `ICLR formatting instructions` throughout the manuscript.
> > > >
> > > > ---
> > > > References
> > > >
> > > > - [1] Jordan et al, Evolving interpretable plasticity for spiking networks, eLife, 2021.
> > > > - [2] Confavreux et al, Memory by a thousand rules: Automated discovery of multi-type plasticity rules reveals variety & degeneracy at the heart of learning, bioRxiv, 2025.
> > > > - [3] Bittner et al, Behavioral time scale synaptic plasticity underlies CA1 place fields, Science 2017.
> > > > - [4] Glickfeld, L. L., Histed, M. H., & Maunsell, J. H. (2013). Mouse primary visual cortex is used to detect both orientation and contrast changes. Journal of Neuroscience, 33(50), 19416-19422.
> > > > - [5] Garrett, M., Manavi, S., Roll, K., Ollerenshaw, D. R., Groblewski, P. A., Ponvert, N. D., ... & Olsen, S. R. (2020). Experience shapes activity dynamics and stimulus coding of VIP inhibitory cells. elife, 9, e50340.
> > > > - [6] Szatmáry, B., & Izhikevich, E. M. (2010). Spike-timing theory of working memory. PLoS computational biology, 6(8), e1000879.
> > > > - [7] Agnes, E. J., & Vogels, T. P. (2024). Co-dependent excitatory and inhibitory plasticity accounts for quick, stable and long-lasting memories in biological networks. Nature Neuroscience, 27(5), 964-974.
> > > > - [8] Mohebi, A., Wei, W., Pelattini, L., Kim, K., & Berke, J. D. (2024). Dopamine transients follow a striatal gradient of reward time horizons. Nature Neuroscience, 27(4), 737-746.

---

### Author Response · Authors · 2025-12-02
**General Response**

Dear AC, SAC, and PC,

First, we would like to extend our special gratitude to you for your time during this difficult period.
We also deeply appreciate the high-quality reviews. Your efforts are highly valued.

---
We are delighted to note that reviewers find that:
- Our paper studies how heterogeneous synaptic plasticity rules drive behavioral learning in a realistic cortical model, which is "important" (Reviewer `N15c`) and "interesting" (Reviewer `Txin`).
- Our method is a "technical tour de force" (Reviewers `N15c`, `XrNa`) and represents the "first large-scale attempt" (Reviewer `t465`) to optimize thousands of possible plasticity rules within a biologically detailed model.
- Our approach "lays the groundwork for the field to embrace such large scale simulations more easily in the future" (Reviewer `N15c`). Our work offers "fresh perspectives" (Reviewer `Txin`) and "insights into how the brain might implement efficient, heterogeneous plasticity" (Reviewer `t465`).
- Our results and analysis are compelling: the discovered plasticity rules are "scale-invariant" (Reviewers `N15c`, `t465`), capable of "few-shot learning" (Reviewer `Txin`), and are both compact and biologically testable (Reviewer `t465`). Our work is "worthy of being shared with the community" (Reviewer `N15c`) and likely to "stimulate further academic discussion" (Reviewer `Txin`).

---
Our requests:

- Given the lack of response from Reviewers `N15c`, `t465` and `Txin`, we would greatly appreciate it if you could review our responses, as we are confident that all concerns have been fully addressed.

- We also kindly invite you to review our response to Reviewer `XrNa`. We respectfully believe there might be a **slight misunderstanding** regarding our focus, which was viewed as *replacing BP-based optimizers for deep learning*. We have clarified that our main goal is **understanding synaptic plasticity mechanisms from a computational neuroscience perspective**. We have addressed all requests raised by the reviewer with extensive additional experiments and clarifications.

---
All concerns from Reviewer `XrNa` have been well addressed, including:

- **Requirement for detailed sensitivity analysis on gradient descent (GD) optimizers**: While the SGD and Adam are merely included for comparison purposes with rigorous hyperparameter tuning already conducted to ensure fairness, we have added 'Appendix G' to provide a detailed sensitivity analysis for Adam and SGD hyperparameters in response to this concern.
- **Requirement for a clearer scientific scope and take-home message**: We have carefully revised the 'Introduction' and 'Conclusion' sections to state the research question and main findings more explicitly.
- **Requirement for detailed analysis of explored plasticity rules**: Added a new 'Discussion' section and 'Appendix F' to provide detailed discussions and analysis of the explored synaptic plasticity rule population.

---
Key revisions based on feedback:

* **Scientific scope & take-home message**: We revised the 'Introduction' and 'Conclusion' sections to articulate the research question and the specific take-home messages clearly. (Suggested by Reviewers `XrNa`, `t465`).

* **New discussion section**: To better separate empirical findings from their interpretation, we moved discussions of functional equivalence, memory emergence, and innate abilities out of the 'Results' section. These topics now reside in a newly added 'Discussion' section, alongside a 'Limitations and Future Work' subsection (Suggested by Reviewers `N15c`, `XrNa`, `t465`).

* **Detailed analysis of explored plasticity rules**:
Added 'Appendix F' to provide a detailed analysis of the characteristics of the explored plasticity rule population, including constraint enforcement dynamics during optimization via three representative explored rule (see **Fig. S8**), parameter distribution among filtered population (see **Fig. S9**), layer-wise dynamics of overall best performing rule (see **Fig. S10**), spike raster plot of the most prevalent reward-free plasticity rule (see **Fig. S11**) and long-term adaptation/generalization ability analysis for best-performing rule (see **Fig. S15**) (Suggested by Reviewers `N15c`, `XrNa`).

* **Merged introduction and related work sections**: We merged the 'Introduction' and 'Related Work' sections to save space in the main text. The comprehensive related work has been moved to 'Appendix B' (Suggested by Reviewers `N15c`, `XrNa`).

* **Sensitivity analysis of GD optimizers**: We added 'Appendix G', which includes a sensitivity analysis for Adam and SGD hyperparameters. **Fig. 14** and **Fig. 15b** illustrate the performance variations across different learning rates and momentum values for Adam and SGD (Suggested by Reviewers `N15c`, `XrNa`).

---
For more details, please refer to our point-by-point responses.

We believe all concerns have been addressed. We sincerely thank you for your consideration.

Best regards,

Authors

---

### Meta-Review · Area_Chair_hLMv · 2026-01-05

**Summary:**

Three out of the four reviewers were positive about this paper prior to the rebuttal period and I assume they would have been positive after the author responses. Reviewer XrNa's primary concerns are (1) the paper is not answering a scientific question (what do we learn from this about the brain?) and (2) the empirical results do not support the paper's claims.

**Reviewer Concerns:**

The authors revised the introduction and discussion to better frame the scientific question and provided mechanistic analyses of the synaptic plasticity rules. The authors provided some further experimental evidence to address Reviewer XrNa's concern about the comparison to Adam; however, I do not think they addressed the concern about the simplicity of the task.

**Reviewer Scores:**

I think the 3 positive reviewers would have maintained their positive scores. I think Reviewer XrNA would have raised their score from a 2 to a 4 since some of their concerns were addressed.

---

### Decision · Program_Chairs · 2026-01-26

Accept (Poster)